# MRAP2 modifies the signaling and oligomerization state of the melanocortin-4 receptor

Iqra Sohail [1,2,3,13], Suli-Anne Laurin [4,13], Gunnar Kleinau[5], Vidicha Chunilal[6], Andrew Morton[7], Alfonso Brenlla[7], Zeynep Cansu Uretmen Kagiali [2], Marie-José Blouin [4], Javier A. Tello [8], Annette G. Beck-Sickinger [9], Martin J. Lohse[1,10,11], Patrick Scheerer [5], Michel Bouvier [4] ✉, Peter McCormick [3,6,12] ✉, Paolo Annibale [1,7] ✉ & Heike Biebermann [2] ✉

The melanocortin-4 receptor is a G protein-coupled receptor and a key regulator of appetite and metabolism. It can interact with the melanocortin-receptor accessory protein 2, a single transmembrane helix protein known to interact with several different G protein-coupled receptors. However, the consequences of this interaction are not completely understood. Here we report that co-expression of melanocortin-receptor accessory protein 2 has multiple effects on the melanocortin-4 receptor: it enhances G protein-mediated signaling and simultaneously impairs β-arrestin2 recruitment and, consequently, internalization. In addition, co-expression of melanocortin-receptor accessory protein 2 leads to an increased number of monomers of melanocortin-4 receptor by disrupting receptor oligomers. A structural homology model of the active state melanocortin-4 receptor – melanocortin-receptor accessory protein 2 – $G\alpha_s$ complex suggests interaction sites that are relevant for receptor activation. Our data indicate that melanocortin-receptor accessory protein 2 is an accessory protein that interacts with and influences melanocortin-4 receptor structure, biasing its signaling towards G protein-mediated effects.

The melanocortin receptor accessory proteins (MRAPs), namely MRAP1 and MRAP2, are single helix membrane-spanning proteins that have been shown to interact with different G protein-coupled receptors (GPCRs)[1], such as the orexin 1 receptor (OXR1)[2], prokineticin 1 receptor (PKR1)[3], and ghrelin receptor (GHSR1a)[4]. It has been hypothesized that MRAPs act in a similar manner to another family of accessory proteins, called receptor activity-modifying proteins (RAMPs), which associate with numerous class B GPCRs[5–11].

[1]Max-Delbrück-Center for Molecular Medicine-Berlin, Berlin, Germany. [2]Charité Universitätsmedizin Berlin, Corporate Member of Freie Universität Berlin and Humboldt-Universität zu Berlin, Institute for Experimental Paediatric Endocrinology, Berlin, Germany. [3]Department of Pharmacology and Therapeutics, University of Liverpool, Liverpool, UK. [4]Institute for Research in Immunology and Cancer, Department of Biochemistry and Molecular Medicine, Université de Montréal, Montréal, QC, Canada. [5]Charité Universitätsmedizin Berlin, corporate member of Freie Universität Berlin and Humboldt-Universität zu Berlin, Institute of Medical Physics and Biophysics, Group Structural Biology of Cellular Signaling, Berlin, Germany. [6]Centre for Endocrinology, William Harvey Research Institute Queen Mary University of London, London, UK. [7]School of Physics and Astronomy, University of St Andrews, St Andrews, UK. [8]School of Medicine, University of St Andrews, St Andrews, UK. [9]Faculty of Life Sciences, Institute of Biochemistry, University of Leipzig, Leipzig, Germany. [10]ISAR Bioscience Institute, Planegg/Munich, Germany. [11]Rudolf Boehm Institute for Pharmacology and Toxicology, University of Leipzig, Leipzig, Germany. [12]XJTLU-University of Liverpool Joint Centre for Pharmacology and Therapeutics, Liverpool, UK. [13]These authors contributed equally: Iqra Sohail, Suli-Anne Laurin. ✉e-mail: michel.bouvier@umontreal.ca; peter.mccormick@liverpool.ac.uk; pa53@st-andrews.ac.uk; heike.biebermann@charite.de

Both MRAP subtypes have been shown to interact with all five members of the melanocortin receptor (MCR) family[12]. MRAP1 is mainly expressed in the adrenal gland and is essential for melanocortin-2 receptor (MC2R) trafficking to the cell surface as well as for adrenocorticotropic hormone (ACTH) binding. MC2R is essentially the sole and indispensable receptor mediating cortisol response to stress[13]. Mutations in MRAP1 as well as in MC2R result in familial glucocorticoid deficiency[14].

Important evidence for direct structural interaction between MRAPs and class A GPCRs was recently provided by the 3D structure of the active MC2R-G protein complex with ACTH and MRAP1, which was solved by cryo-electron microscopy (cryo-EM)[15]. Functional studies indicated a reduction of MC4R signaling by MRAP1[16].

MRAP2 is predominantly expressed in tissues impacting metabolism and glucose homeostasis, including the central nervous system (CNS), the pancreas, the gut and adipose tissues[17]. Within the mouse CNS, especially in the hypothalamus and in the paraventricular nucleus (PVN), MRAP2 is co-expressed in some neurons together with the MC4R, where it has been proposed to be important for some MC4R functions[4,18–20]. MRAP2 and MC4R messenger RNAs were both found in human PVN (The Human Protein Atlas) at a ratio around 3:1. The PVN is a hypothalamic region critical for energy homeostasis and the MC4R is a crucial player in energy metabolism[21,22]. For that reason, the MC4R is one of the prime targets for potential pharmacological regulation of appetite and body weight[23], as witnessed by more than 160 different pathogenic human variants causing obesity[24].

In addition to rare human heterozygous *MRAP2* mutations that result in increased body weight, global and brain specific *Mrap2* deletion in mice leads to marked obesity[4,25]. Moreover, a large-scale study in pigs identified *MRAP2* as a candidate gene associated with daily weight gain, supporting its importance for energy balance in different species[26]. This effect on body weight regulation is further underpinned by a previous study on the role of MRAP2 in zebrafish feeding and growth[27]. In addition to its canonical Gs protein-dependent cell signaling, the MC4R activates the Gq/11 family signaling pathway, which is also involved in appetite regulation[28]. Agonist-stimulation further triggers β-arrestins recruitment to the receptor to terminate G-protein

signaling at the plasma membrane and to initiate receptor endocytosis[29].

Recent investigations suggested that MRAP2 and MC4R co-expression led to an increased signaling[27,30]. It was hypothesized that the observed increase in Gs-mediated cAMP signaling could be related to MC4R homodimer separation[30]. This scenario, however, has never been explored in detail experimentally, albeit MC4R homodimerization has been reported by us and others, both for human MC4R and in orthologous forms such as *Xiphophorus* MC4R[29,31–33].

Many questions regarding the mode of action and the resulting effects of the interaction between MRAP2 and MC4R are thus still unclear. In this study, we aimed to investigate the effect of MRAP2 on different MC4R signaling pathways and supramolecular organization, using a suite of fluorescence and bioluminescence resonance energy transfer (FRET and BRET)-based assays, single cell imaging and functional assays[34]. Collectively, our data point to an important role of MRAP2 in modulating the supramolecular organization state of MC4R, namely its increased monomerization, while directly impacting receptor signaling.

## Results
### MRAP2 expression increases Gs-mediated cAMP accumulation in the basal and ligand-activated states of MC4R

To investigate the effect of MRAP2 on MC4R signaling, we first examined the effect of MRAP2 expression on the MC4R-induced Gs activation in heterologously transfected HEK293 cells. Using the Epac-S[H187] FRET biosensor[35], the increase in cAMP formation was determined as a reduction in FRET between a cyan fluorescent donor and a yellow fluorescent acceptor, and plotted as normalized FRET responses upon stimulation with the agonist α-MSH in the presence or absence of varying ratios of MC4R:MRAP2. Our data show that upon co-expression of either the same DNA mass ratio of MC4R and MRAP2 (1:1, indicated as 1 + 1), or a fourfold excess of MRAP2 to MC4R (1:4 ratio, indicated as 1 + 4), there is a leftward shift of the average concentration-response curves for α-MSH-stimulated cAMP production (Fig. 1A), reflecting an increased potency of the agonist for the MC4R promoted cAMP production. Our selection of the 1 + 1 and 1 + 4

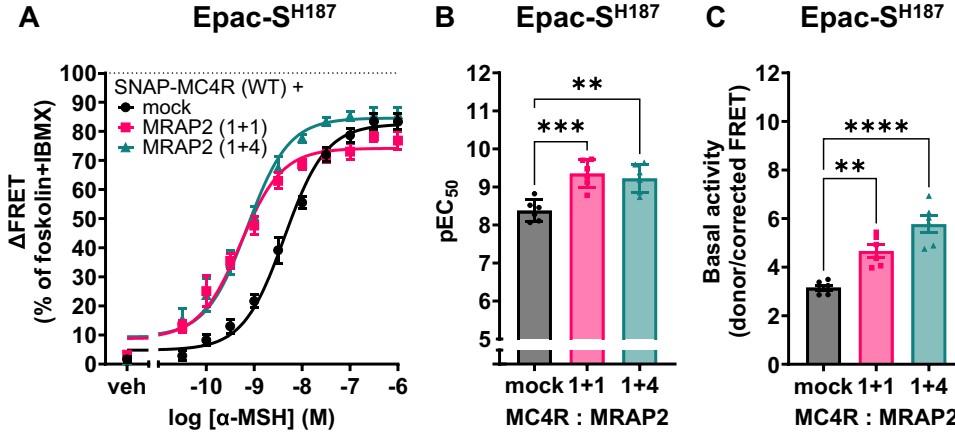

**Fig. 1 | MRAP2 co-expression enhances cAMP signaling. A** Average concentration-response curves of cAMP response upon MC4R stimulation with α-MSH, with and without different amounts of MRAP2 in transiently transfected HEK293 cells using the EpacS[H187] biosensor. Data are normalized to forskolin response in the presence of the phosphodiesterase inhibitor IBMX (100%). MC4R is first expressed with an equivalent amount of pcDNA (mock). Plasmid DNA coding for MRAP2 is then co-expressed either in equivalent amounts (1 + 1) or with a fourfold higher molar concentration with respect to the MC4R (1 + 4), always maintaining constant the total amount of DNA transfected. Mean and standard error of the mean (± SEM) of six independent experiments (with three technical replicates each) are shown. **B** pEC$_{50}$ values obtained using the Epac-S[H187] FRET

biosensor. Results are represented in a bar chart as mean ± SEM, in which individual pEC$_{50}$ data point from the six individual experiments are shown as overlaid dot plot. One-way ANOVA as statistical test was performed with Tukey's multiple comparisons post-hoc test (**: adjusted *p*-value = 0.0018, ***: adjusted *p*-value = 0.0005). **C** Basal cAMP values (as assessed by FRET ratio) in the presence or absence of MRAP2 measured using the Epac-S[H187] FRET biosensor in cells transfected different amounts of MRAP2 relative to the MC4R. Results are represented in a bar chart as mean ± SEM, in which individual data point from the six individual experiments are shown as overlaid dot plot. One-way ANOVA as statistical test was performed with Dunnett's multiple comparisons post-hoc test (**: adjusted *p*-value = 0.0014, ****: adjusted *p*-value < 0.0001).

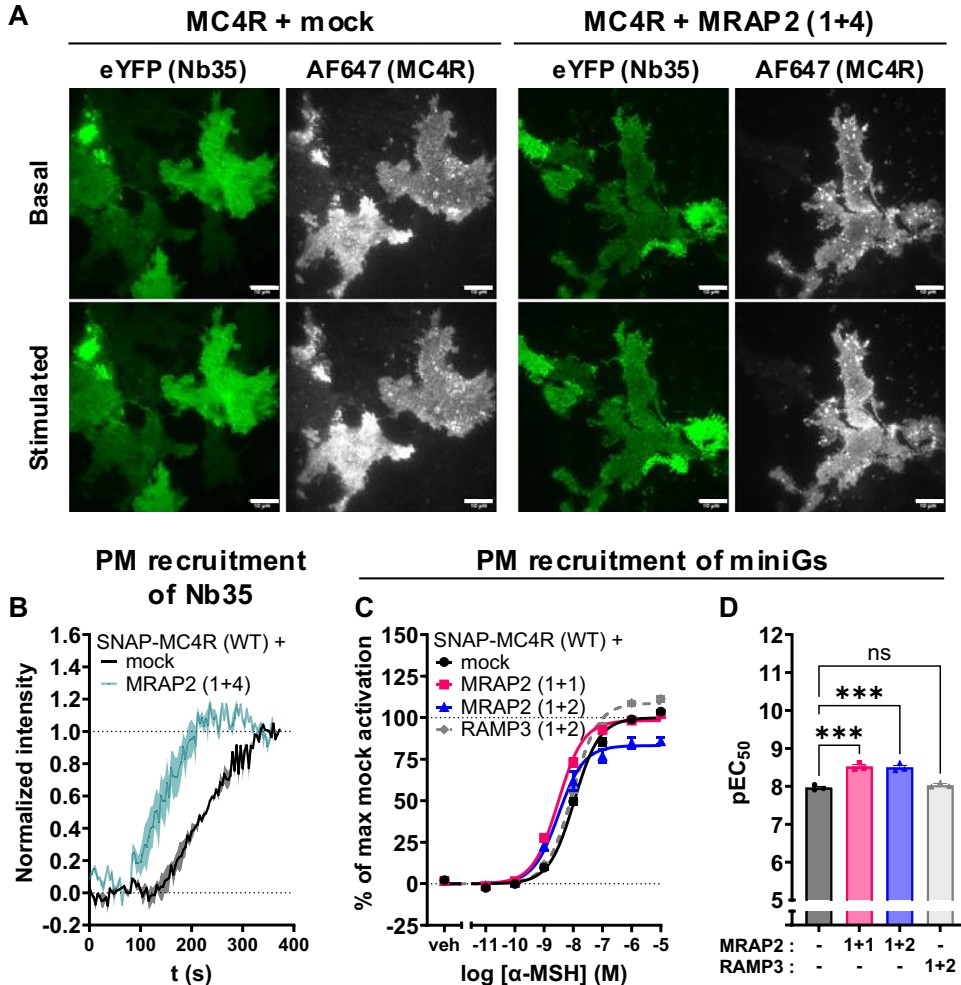

**Fig. 2 | MRAP2 enhances Gα_s activation by the MC4R. A** Representative images from TIRF microscopy of the basolateral membrane of HEK293AD cells expressing SNAP-tagged MC4R labeled using Alexa Fluor 647 (gray) and Nb35-eYFP (green). Images were taken pre- (top) and 4 min post- (bottom) stimulation with 1 μM NDP-α-MSH for cells expressing SNAP-MC4R alone or with MRAP2 (at a ratio of 1:4) and indicate recruitment of Nb35 (green intensity). Scale bar is 10 μm. **B** Kinetics of recruitment of Nb35 to the SNAP-MC4R–Gs complex, without (black; 3 transfections, 27 single cells) and with MRAP2 (1 + 4) (turquoise; 3 transfections, 42 cells). Shown are the normalized mean fluorescence intensities recorded at the membrane in the eYFP channel; the shaded area represents ± SEM from 3 individual experiments. **C** Average concentration-response curves of Rluc8-miniGs recruitment to the plasma membrane localization sensor rGFP-CAAX upon MC4R stimulation with α-MSH for 45 min, in the presence or absence of MRAP2 or RAMP3, at the indicated ratio with respect to MC4R, in transiently transfected HEK293-SL cells. Normalized data are expressed as mean ± SEM of three independent experiments. **D** pEC_50 values from the miniGs recruitment BRET experiments. Results are represented in a bar chart as mean ± SEM, in which individual pEC_50 data point from the three individual experiments are shown as overlaid dot plot. Statistical analysis was performed using ordinary one-way ANOVA with Dunnett's multiple comparisons post-hoc test (***: adjusted *p*-value = 0.0001).

ratios stems from the reported observation that mRNA levels in the paraventricular hypothalamic nucleus are of the order of 1 + 2 (The Human Protein Atlas)[36] (Supplementary Table 1). Expression ratios of 1 + 1 and 1 + 4 were therefore chosen as boundaries encompassing this reported ratio.

Interestingly, both ratios lead to a similar shift in the EC_50 obtained from fitting the concentration-response curves using Hill's equation, namely from 4.5 nM to 1.3 nM, as seen in Fig. 1B, suggesting a saturable effect. We then determined basal (non-ligand stimulated) cAMP concentrations, in cells transfected with both MRAP2 and the MC4R, as opposed to cells transfected with the MC4R and mock DNA, to keep the DNA content stable. As shown in Fig. 1C, a statistically significant increase in basal activity occurs when MRAP2 DNA is transfected in 1 + 1 or 1 + 4 MC4R to MRAP2 DNA ratios. Results are expressed as donor/FRET ratio, which increases proportionally to cAMP formation.

## MRAP2 expression favours Gs recruitment to the MC4R and may enhances α-MSH binding

To confirm whether the observations reported in Fig. 1A−C can be ascribed to enhanced recruitment of the stimulatory heterotrimeric Gs protein at the agonist-bound MC4R, we exploited the nanobody 35 (Nb35), which recognizes the active conformation of Gα_s, and the miniGs, which binds the active conformation of Gs-coupled GPCR.

In our first assay, a fluorescently labeled Nb35 (Nb35-eYFP), which is known to bind with high affinity the active conformation of the Gα_s subunit[37,38], was transfected in HEK293 cells co-expressing SNAP-MC4R (Fig. 2A). The addition of an N-terminal SNAP-tag does not affect MC4R downstream signaling (Supplementary Fig. 1). Total internal fluorescence microscopy (TIRF) was then used to monitor the nanobody recruitment at the plasma membrane of MC4R-expressing cells upon stimulation with 1 μM of NDP-α-MSH. This agonist was employed to maximize the signal-to-noise ratio in these assays, since it is 30 times

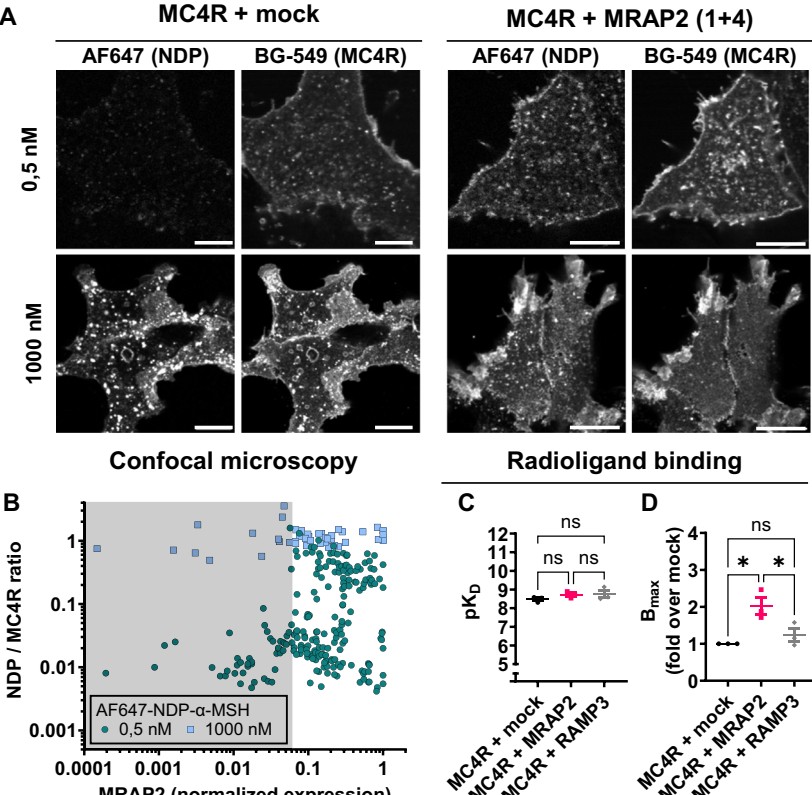

**Fig. 3 | Binding of NDP-α-MSH is increased in HEK293 expressing MRAP2.**
**A** Confocal maximum intensity projections of representative HEK293 cells trans-fected with either SNAP-MC4R + mock or SNAP-MC4R + MRAP2 (1 + 4), seeded on the same coverslip and then labelled using SNAP-surface549. The cells were then incubated for 5 min with either 0.5 nM or 1000 nM Alexa647-NDP-α-MSH and imaged. Contrast is set to the same values in each channel. Maximum intensity projections were constructed in Fiji before running Cellpose-based cell segmen-tation to extract individual cells intensity values. Scale bar is 10 μm. **B** Chart of the normalised NDP-α-MSH ligand signal (to SNAP-MC4R receptor expression) per cell for 0.5 nM ligand (turquoise circles, $n = 5$ independent transfections, 226 cells) and 1000 nM ligand (green square, $n = 2$ independent transfections, 45 cells). Shading

indicates MRAP2 positive cells, as determined by visual inspection of confocal micrographs. **C** Affinity and **D** maximal binding parameters extracted from saturation radioligand binding experiments using $[I]^{125}$-NDP-α-MSH on membrane preparations of HEK293-SL cells transfected with MC4R and MRAP2 or RAMP3 at 1 + 1 ratio. Results are represented as mean ± SEM, in which individual data point from the three individual experiments are shown. Results show an increase in $B_{max}$ for MRAP2 and not from the negative control RAMP3, but no change in affinity ($pK_D$) for both conditions. Statistical analysis was performed using ordinary one-way ANOVA with Tukey's multiple comparisons post-hoc test (* = $p < 0.05$, adjusted $p$-value mock vs MRAP2 = 0.0125 and adjusted $p$-value MRAP2 vs RAMP3 = 0.0377). Non-significant comparison is not shown in (**D**).

more potent than α-MSH[39]. Cells were co-transfected with a plasmid coding for Gα$_s$ protein to ensure sufficient Gα$_s$ in the system to allow detection of the 1:1 stoichiometric recruitment of Nb35 to the recep-tors. Expression levels of Gα$_s$ were also confirmed to be stable in the presence or absence of MRAP2 with a fluorescently tagged version of Gα$_s$ (Supplementary Fig. 3). Figure 2A illustrates the enhancement of Nb35 fluorescence (green) at the plasma membrane of SNAP-MC4R-expressing cells (gray) upon agonist addition, in the presence or absence of MRAP2. Figure 2B illustrates the kinetics of Nb35 recruit-ment to the membrane in both cases. Notably, MRAP2 significantly increased the kinetics of the nanobody recruitment, effectively dou-bling the kinetic on-rate, suggesting that Gα$_s$ recruitment to the receptor and its activation are indeed favored by the action of MRAP2.

In the second assay, MC4R activation at the plasma membrane (PM) was confirmed by the increase in the enhanced bystander BRET (ebBRET) signal between Rluc8-miniGs, which recognizes the active form of Gα$_s$-coupled GPCR such as MC4R, and a *Renilla* green fluor-escent protein (rGFP) targeted to the plasma membrane with the CAAX box of KRas[40]. The half log-unit leftward shift of the concentration-response curve upon equivalent transfection of MC4R and MRAP2 (1 + 1) relative to the control mock conditions (Fig. 2C) confirms that

MRAP2 expression favors Gα$_s$-coupling (Fig. 2D). RAMP3, for which no prior indication of an interaction with MC4R has been reported, was used as a negative control, and it indeed showed no effect on miniGs recruitment[41]. In side-by-side experiments, there is also no change in BRET upon stimulation of cells not transfected with MC4R, showing the response is indeed dependent on this receptor (Supplemen-tary Fig. 4A).

These results are mirrored when measuring ligand binding. Using a single cell-based fluorescence assay, we could observe that intact cells transfected with MRAP2 (1 + 4) bind significantly more fluores-cently tagged ligand (AF647-NDP-α-MSH) than the mock transfected counterparts (Fig. 3A). Furthermore, the number of AF647-NDP-α-MSH molecules bound, normalized by the number of receptors expressed, increases as a function of the expression level of MRAP2 in transiently transfected cells (Fig. 3B), in the presence of 0.5 nM AF647-NDP-αMSH. However, at 1000 nM of AF647-NDP-αMSH, binding appears inde-pendent of MRAP2 expression level. When conducting radioligand binding on membranes from HEK293 cells using $[I]125$-NDP-α-MSH, we observed an MRAP2-independent affinity (Fig. 3C). On the other hand, Bmax values were increased in membranes from cells transfected with MRAP2 (1 + 1), suggesting an increase of the number of receptors present at the plasma membrane (Fig. 3D).

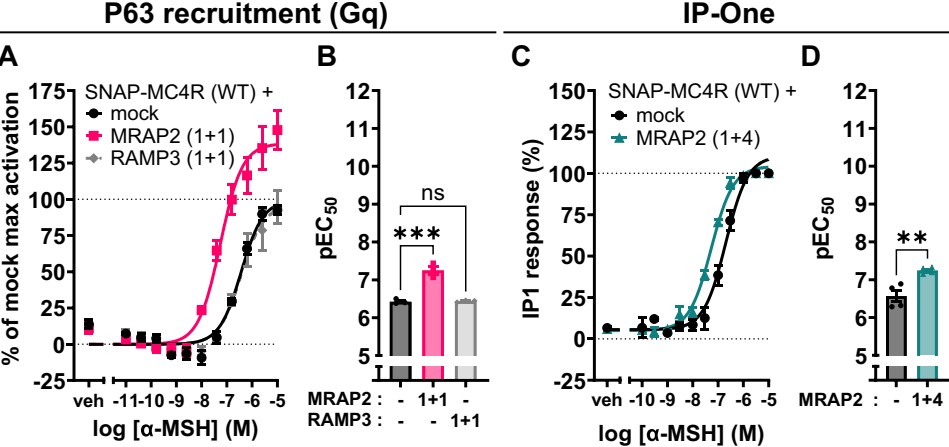

**Fig. 4 | MRAP2 enhances Gq activation. A** Mean concentration-response curves of the Gq-family selective effector p63RhoGEF-RlucII recruitment to the plasma membrane localization sensor rGFP-CAAX upon MC4R stimulation with α-MSH for 45 min, with overexpression of Gq in the presence or absence of MRAP2 or RAMP3 in transiently transfected HEK293-SL cells. MRAP2 or RAMP3 plasmid DNA is co-transfected with the same amount (1:1) with respect to MC4R DNA. Normalized data are expressed as mean ± SEM of three independent experiments. **B** $pEC_{50}$ from the p63RhoGEF recruitment BRET experiments for Gq. Results are represented in a bar chart as mean ± SEM, in which individual $pEC_{50}$ data point from the three individual experiments are shown as overlaid dot plot. Statistical analysis was performed using ordinary one-way ANOVA with Dunnett's multiple comparisons post-hoc test (***: adjusted *p*-value = 0.0002). **C** Average concentration-response curves of IP-One assay upon MC4R stimulation using α-MSH, with and without MRAP2 1 + 4 DNA ratio with respect to MC4R, in transiently transfected HEK293T cells. Data are expressed as mean ± standard error of the mean (± SEM) of four independent experiments. **D** $pEC_{50}$ values across all the replicates of the IP-One concentration-response curves for the two conditions. Results are represented in a bar chart as mean ± SEM, in which individual $pEC_{50}$ data point from the four individual experiments are shown as overlaid dot plot. Statistical analysis was performed using an unpaired two-tailed *t*-test (**: adjusted *p*-value = 0.005).

## MRAP2 increases MC4R-induced activation of Gq signaling pathway

Based on these observations, we further investigated the signal transduction by another G protein that has been shown to be engaged by MC4R, namely $G\alpha_q$, using two different sets of assays.

First, we exploited ebBRET between p63RhoGEF, an effector that selectively interacts with activated Gα subunits of the Gq/11 subfamily, and plasma membrane-targeted rGFP[42]. Upon co-expression with MRAP2, we observed a leftward shift of the concentration-response curve upon stimulation with the agonist α-MSH, as well as an increase in the maximal response ($E_{max}$) (Fig. 4A). $EC_{50}$ values were left shifted by almost one log-unit (Fig. 4B). In side-by-side experiments, RAMP3, used as a negative control, had no impact on the α-MSH-promoted response. No response was observed after α-MSH challenge in the absence of SNAP-MC4R expression, showing that the response is indeed MC4R-dependent (Supplementary Fig. 4D). The p63RhoGEF sensor is also unresponsive in cells not overexpressing $G\alpha_q$, both in the presence or absence of MC4R, showing that the observed increase in BRET is solely dependent of $G\alpha_q$ and no other G protein (Supplementary Fig. 4C).

Second, we measured second messenger production downstream of $G\alpha_q$ activation. Using IP-One HTRF assay, which monitors inositol monophosphate (IP1) accumulation, a metabolite of inositol-1,4,5-trisphosphate (IP3), this latter being produced following phospholipase C (PLC) stimulation downstream of $G\alpha_q$ activation. We found that MRAP2 co-expression with MC4R yields a leftward shift in the concentration-response curve of α-MSH (Fig. 4C). This MRAP2-promoted increase in Gq activation is also supported by NFAT-reporter assays, as a read-out of calcium production, which is typically associated with PLC activation downstream of Gq activation (Supplementary Fig. 2). Overall, co-expression of MRAP2 in a 1 + 4 amount with respect to the receptor, yields a threefold decrease in $EC_{50}$ of IP1 production (Fig. 4D). The observation that, in contrast to the p63RhoGEF BRET assay, no increase in the maximal response was observed most likely reflects a saturation of the IP1 response at the level of receptor expression, as the response was within dynamic range of the standard curve.

## MRAP2 modulates MC4R constitutive and ligand-dependent internalization

It is becoming clear that changes in timing and location of GPCR signaling can alter cell response and physiology[43]. The observation that MRAP2 has a role in modulating the downstream signaling of MC4R suggests that one potential mechanism for MRAP2 to achieve this modulation might be to influence trafficking of MC4R. This question motivated us to investigate the role of MRAP2 expression in the localization and trafficking of MC4R.

First, we conducted a direct inspection of receptor localization by means of confocal microscopy. Labeling was achieved by a membrane-impermeable dye (SNAP-Surface Alexa Fluor 647) that irreversibly binds to SNAP-MC4R receptors on the plasma membrane. We then performed a pulse chase experiment that allowed us to monitor any labeled SNAP-MC4R that internalized during a 30-min incubation period. Next, we compared the cellular distribution of the transfected SNAP-MC4R in HEK293 cells, in the presence or absence of MRAP2 co-expression (1 + 4). Fig. 5A displays representative confocal micrographs: when SNAP-MC4R is expressed alone, considerable intracellular fluorescence is evident, due to receptor constitutive internalization during the incubation with the SNAP-tag dye. Notably, the intracellular signal largely disappears when co-expressed with MRAP2. In this case the expression pattern mimics the one observed upon preincubation for 30 min with 100 nM AgRP, an inverse agonist that should block constitutive internalization[44]. The use of AgRP in addition to MRAP2 co-expression did not significantly alter this pattern. Thus, it appears that MRAP2 effectively reduces constitutive MC4R internalization.

We then investigated MC4R internalization upon agonist stimulation. In HEK293T cells co-expressing SNAP-MC4R and the endosomal marker Cerulean-Rab5 (Fig. 5B, **top**), we counted the number of endosomes within the cell before stimulation and 10 min after the addition of the agonist NDP-α-MSH at saturating concentrations (1 μM). In both scenarios, MRAP2 expression decreases significantly the number of endosomes observed per cell (Fig. 5B, **bottom**), as quantified in Fig. 5D. Upon co-expression of MRAP2, there was no effect of NDP-α-MSH on the number of MC4R-positive endosomes per cells,

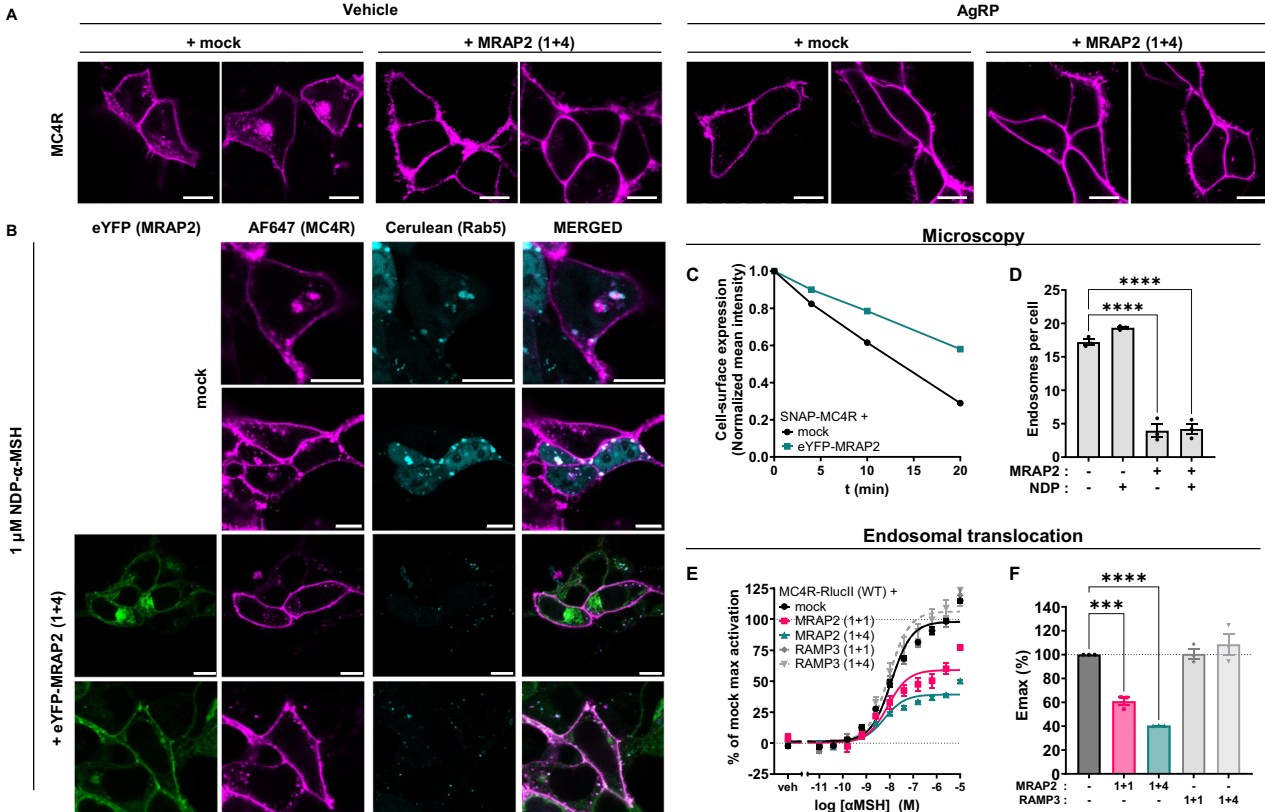

**Fig. 5 | MRAP2 decreases constitutive and ligand-dependent endocytosis of MC4R. A** Representative confocal micrographs of the localization and trafficking of SNAP-tagged MC4R in basal conditions or after the addition of 100 nM of the inverse agonist AgRP for 30 min in HEK293T cells in absence of presence of co-expression of MRAP2 (1 + 4). More than 3 independent experiments have been done. Scale bar is 10 μm. **B** Internalization of SNAP-tagged MC4R after 30 min incubation using NDP-α-MSH (1 μM), co-expressed with the endosomal marker Cerulean-Rab5, with or without co-expression of eYFP-MRAP2 (3 transfections, 14 cells). Scale bar is 10 μm. **C** Time evolution of SNAP-MC4R intensity at the plasma membrane of cells expressing SNAP-MC4R alone or in co-expression with eYFP-MRAP2 (1 + 4) upon stimulation with NDP-α-MSH (1 μM). Mean values ± 95% CI of three independent experiments are indicated. **D** Quantification of the number of endosomes per cell detected using Cerulean-Rab5 staining with and without co-expression of MRAP2 (1 + 4) together with MC4R, both in basal and agonist stimulated conditions. Results are represented in a bar chart as mean ± SEM, in which individual points represent replicates from the three independent experiments are

shown as overlaid dot plot. Statistical analysis was performed using ordinary one-way ANOVA with Dunnett's multiple comparisons post-hoc test (****: adjusted $p$-value < 0.0001). **E** Mean concentration-response curves of the translocation of MC4R-RlucII to the early endosome localization sensor rGFP-FYVE upon stimulation with α-MSH for 1 h, in the presence or absence of MRAP2 or RAMP3 in transiently transfected HEK293-SL cells. MRAP2 or RAMP3 plasmid DNA is co-transfected with the same amount (1 + 1) or fourfold the amount (1 + 4) of the MC4R-RlucII plasmid DNA. Normalized data are expressed as mean ± SEM of three independent experiments. **F** $E_{max}$ values from the translocation to early endosome BRET experiments. Results are represented in a bar chart as mean ± SEM, in which individual Emax data point from the three individual experiments are shown as overlaid dot plot. Statistical analysis was performed using ordinary one-way ANOVA with Dunnett's multiple comparisons post-hoc test (***: adjusted $p$-value = 0.0005; ****: adjusted $p$-value < 0.0001). Non-significant pairwise comparison are not shown.

suggesting an effective impairment of receptor internalization by MRAP2. This is paralleled by a reduction in the rate of receptor removal from the membrane upon agonist stimulation, illustrated in Fig. 5C. Additional examples are shown in Supplementary Fig. 5.

These results were complemented by an internalization read-out, exploiting ebBRET between MC4R-RlucII and an acceptor rGFP fused to the FYVE zinc-finger domain of endofin, which acts as an early endosome targeting sequence (Fig. 5E)[45]. This approach, in which the addition of the RlucII in C-terminal of the MC4R does not perturb its function (Supplementary Fig. 1C) allowed us to conduct a concentration-response readout of MC4R internalization, comparing the effect of co-transfecting increasing amounts of MRAP2. Upon MRAP2 overexpression, either at a 1 + 1 or 1 + 4 ratio, there is a clear decrease in the BRET signal within the endosomes (Fig. 5F), again suggesting an impaired ligand-dependent internalization of MC4R in the presence of MRAP2. This is in stark contrast with measurements conducted using RAMP3 as a negative control, where this unrelated accessory protein has no effect on the internalization of MC4R.

Interestingly, when monitoring trafficking and cellular localization of MC4R in N7 murine embryonic hypothalamic cell line by confocal microscopy, these effects appeared to be amplified. N7 cells labeled with membrane impermeable SNAP-Surface® Alexa Fluor® 647 and expressing SNAP-MC4R, together with a cytosolic reporter, displayed negligible plasma-membrane localization of the receptor (Supplementary Fig. 6A). On the other hand, those cells expressed very well on the membrane the SNAP-β1-adrenoreceptor (SNAP-β1AR), another GPCR, used here as a positive control (Supplementary Fig. 6B).

When MC4R was co-transfected together with MRAP2 (1 + 1), it displayed a clear membrane localization, suggesting an even more prominent role of MRAP2 in regulating the trafficking of the receptor in this cell line in comparison to HEK293T cells consistent with its role in inhibiting constitutive endocytosis (Supplementary Fig. 6C).

### Overexpression of MRAP2 decreases β-arrestin2 recruitment to the receptor

These results point to both enhanced Gs- and Gq/11-dependent downstream signaling, but, at the same time, a reduced trafficking and

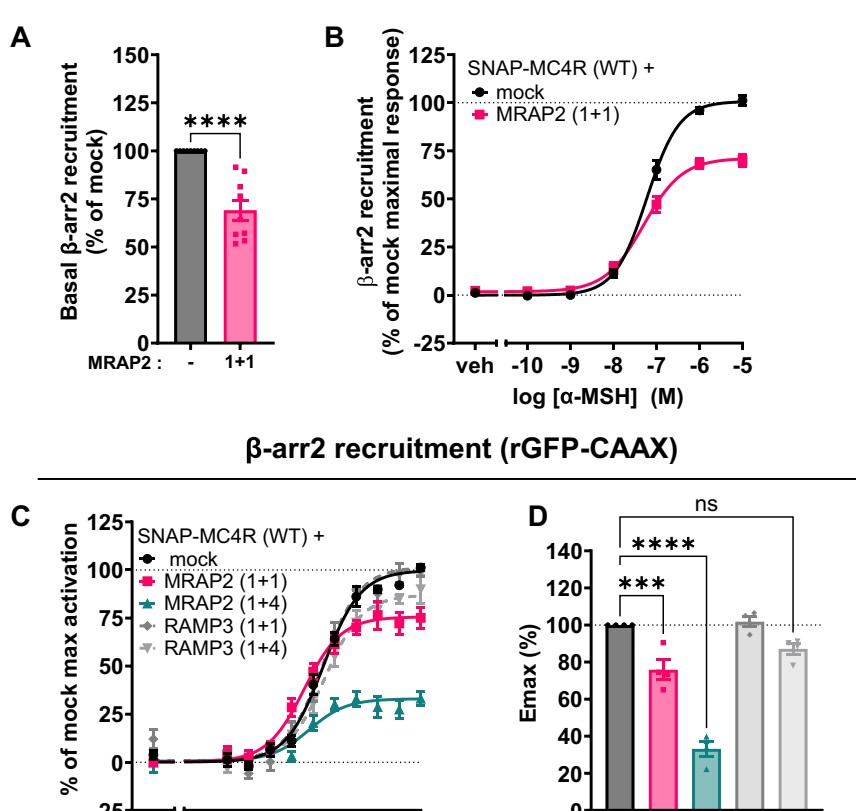

**Fig. 6 | β-arrestin2 recruitment to MC4R in the presence or absence of MRAP2.**
**A** Bioluminescence-complementation based readout of SmBiT-β-arrestin2 recruitment to MC4R-LgBiT under basal conditions, with and without co-expression of MRAP2 (1 + 1). Data are expressed as percentage of the basal β-arrestin recruitment to the MC4R-WT in absence of MRAP2 (mock). Results are represented in a bar chart as mean ± SEM, in which individual points represent average values from the nine independent experiments, are shown as overlaid dot plot. Statistical analysis was performed with two-tailed unpaired *t*-test (****: adjusted *p*-value < 0.0001). **B** Average concentration-response curve of SmBiT-β-arrestin2 recruitment to MC4R-LgBiT upon α-MSH stimulation with and without co-expression of MRAP2 (1 + 1). Data represent mean ± SEM from three independent experiments with four replicates each. EC$_{50}$ values and normalized maximal recruitment derived from curve fitting were 100% and 60 nM ± 8 nM for control condition and 67% and 50 nM ± 8 nM with overexpression of MRAP2 (1 + 1). **C** Mean concentration-response curves of β-arrestin2-RlucII recruitment to the plasma membrane

localization sensor rGFP-CAAX upon MC4R stimulation with α-MSH for 2 min, in the presence or absence of MRAP2 or RAMP3 in transiently transfected HEK293-SL cells. MRAP2 or RAMP3 plasmid DNA is co-transfected with the same amount (1 + 1) or fourfold the amount (1 + 4) of the SNAP-MC4R plasmid DNA (140 ng), with total amount of DNA being constant. EC$_{50}$ values were 58 ± 10 nM, 21 ± 4 nM, and 32 ± 8 nM, respectively in the presence or absence of 1 + 1 or 1 + 4 MRAP2, and were not significantly different. Normalized data are expressed as mean ± SEM of four independent experiments. **D** E$_{max}$ values from the β-arrestin2 recruitment BRET experiments. Results are represented in a bar chart as mean ± SEM, in which individual Emax data point from the four individual experiments are shown as overlaid dot plot. Statistical analysis was performed using ordinary one-way ANOVA with Dunnett's multiple comparisons post-hoc test (***: adjusted *p*-value = 0.0006; ****: adjusted *p*-value = <0.0001). Non-significant pairwise comparisons are not all shown.

internalization rate. The latter led us to investigate if β-arrestin2 recruitment to the receptor, a canonical step following receptor activation that allows for clathrin-mediated endocytic internalization of the receptor for recycling and/or degradation, was affected by co-expression of MRAP2. We employed a luminescence-based protein complementation assay (NanoBiT), whereby luminescence from a complemented luciferase protein is observed upon close proximity of two proteins. In this case, the MC4R is fused with LgBiT at its C-terminus, whereas the β-arrestin2 is fused at its N-terminal with the SmBit[46].We observed that overexpression of MRAP2 at 1 + 1 ratio already leads to a significant decrease in β-arrestin2 recruitment directly to the receptor (Fig. 6A). This reduction of β-arrestin2 recruitment in the basal state parallels the data of reduced constitutive internalization observed in our confocal microscopy quantification (Fig. 5A). Furthermore, α-MSH stimulation results in reduced β-arrestin2 recruitment in the presence of MRAP2 (Fig. 6B), mirroring

the reduction of agonist-dependent MC4R internalization observed in the presence of MRAP2 (Fig. 5B). These results are again recapitulated in an ebBRET assay conducted using a C-terminally labeled β-arrestin2-RlucII and the membrane targeted rGFP-CAAX (Fig. 6C)[45]. Here, for an equivalent amount of MRAP2 and MC4R transfected (1 + 1) and a fourfold excess of MRAP2 (1 + 4), we observed a substantial drop in β-arrestin2 recruitment E$_{max}$ up to almost 40% whereas no change was observed upon co-expression of the negative control RAMP3. No change in BRET was detected in absence of SNAP-MC4R over-expression, showing the recruitment at the plasma membrane driven by MC4R (Supplementary Fig. 4B).

These data suggest that the molecular mechanism of the interaction between MRAP2 and MC4R favors Gs- and Gq/11-coupling, while interfering with internalization, as well as β-arrestin2 engagement.

Despite the possible link between the increase in G protein signaling and the decrease β-arrestin2 recruitment and receptor

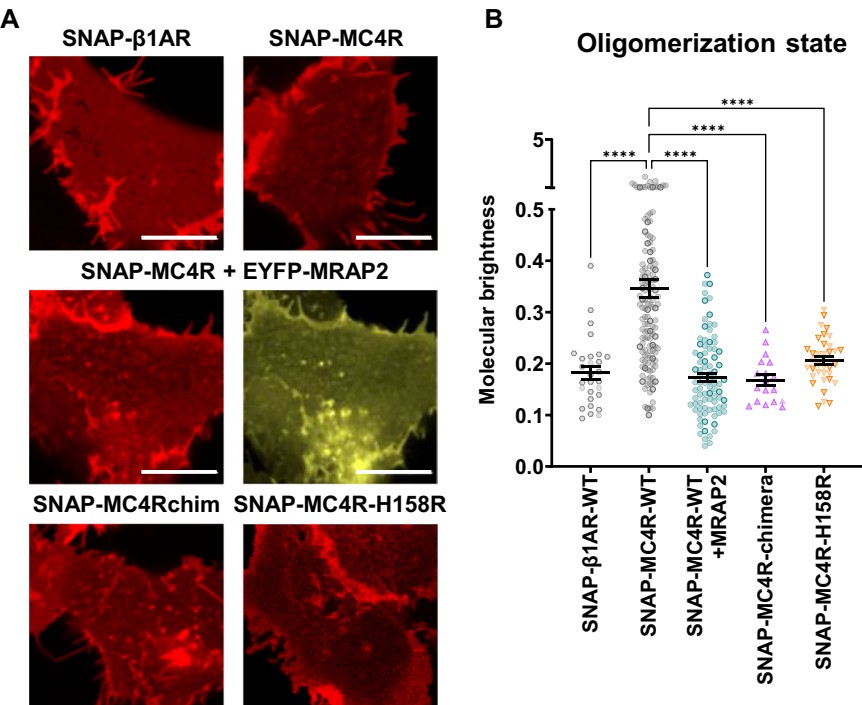

**Fig. 7 | Oligomeric state of MC4R in the presence or absence of MRAP2.**
**A** Representative confocal micrographs of the basolateral membrane of HEK293 cells expressing SNAP-tagged GPCRs, all labeled using SNAP-surface647. From the top left the SNAP-β1AR ($n = 4$ transfections, 28 cells), SNAP-MC4R ($n = 6$ transfections, 135 cells), SNAP-MC4R in combination with eYFP-labeled MRAP2 (as a reference), SNAP-MC4R in presence of untagged MRAP2 ($n = 4$ transfections, 85 cells), SNAP-MC4Rchim ($n = 3$ transfections, 18 cells) and finally the SNAP-MC4R-

H158R ($n = 3$ transfections, 35 cells). Scale bar is 10 μm. **B** Molecular brightness values obtained for each of the control groups listed in (**A**). Here, MRAP2 used to generate the micrographs of the SNAP-MC4R-WT + MRAP2 1:4 condition is untagged. Each dot represents the mean value from multiple regions of interest (ROIs) within a single cell. Several ROIs per cell were imaged. One-way ANOVA has been performed with Dunnett's multiple comparisons post-hoc test (****: adjusted $p$-value < 0.0001). Mean ± SEM are shown in black.

endocytosis, we shall point out that the MRAP2-mediated increase in G protein signaling cannot be solely due to a reduction in β-arrestin-promoted endocytosis, since a leftward shift in Gs activation was also observed in β-arrestin1 and β-arrestin2 double knock-out cells when transfected with MRAP2 (Supplementary Fig. 7).

## MRAP2 expression alters the MC4R oligomeric equilibrium towards a monomeric state

Receptor oligomerization has long been a mechanism invoked to explain the modulation of GPCR downstream signaling, although less commonly this has been shown to influence the ligand-dependent response or to be in turn influenced by ligands[47–49]. Based on BRET-based studies, we have previously suggested that the MC4R has a homodimeric arrangement[31], which can impact downstream signaling of the receptor, in particular coupling to Gs and Gq. RAMP

Thus, we tested the hypothesis that co-expression of MRAP2 can modulate the oligomeric state of MC4R. For this purpose, we performed a single cell assay based on molecular brightness, similar to that used for other GPCRs[50]. We first conducted confocal microscopy of the basolateral membrane of cells expressing different receptor N-terminally tagged with SNAP-tag (Fig. 7A). The brightness mean of the SNAP-β1AR was identified as a monomeric reference (Supplementary Fig. 8), based on previous observation using this experimental method[50]. The fingerprint of the SNAP-MC4R, yielding a brightness approximately 1.7 times larger than a monomer, indicates a coexistence of higher-order oligomeric species, averaging on an approximately dimeric condition (Fig. 7B). However, co-expression of MRAP2 (1 + 4) leads to a change in the MC4R oligomeric fingerprint, and shifts the equilibrium towards a more monomeric population. Furthermore, MRAP2 appears to increase the average number of cell surface-expressed MC4Rs in a concentration-dependent way (Supplementary

Fig. 9A), leading to a significantly increased receptor expression levels in cells co-expressing MRAP2 (1 + 4) (Supplementary Fig. 9B), in line with the observations reported in Figs. 3C and 5.

Based on these results, we validated the direct interaction between MC4R and MRAP2 using a related fluorescence fluctuation spectroscopy method to Molecular Brightness, namely Fluorescence Cross-Correlation Spectroscopy (FCCS) (ref. 51, and reviewed in ref. 34). FCCS data shows unambiguously that SNAP-MC4R co-diffuses with EYFP-MRAP2 (Supplementary Fig. 10), in a fashion comparable to the constitutively dimeric CXCR4[47]. On the contrary, a negative control monitoring EYFP-MRAP2 co-transfected with the SNAP-β1AR, did not display any co-diffusion.

## Signaling of homodimerization-impaired MC4R variants is still affected by MRAP2

Our data delineate a scenario in which the MC4R is predominantly expressed as a dimer that will disassemble into a monomer upon interaction with MRAP2. This is accompanied by an increase in Gs and Gq activation, an elevated downstream cAMP and IP3 production, a decrease in β-arrestin2-mediated recruitment, and a significant reduction in receptor internalization.

This raises the question as to whether MC4R monomerization influences the receptor towards G-protein coupling, including both Gs and Gq, as opposed to β-arrestin2, or rather if the observed changes in signaling arise from a direct modulation of MC4R protomers by MRAP2.

We tested this hypothesis by exploiting two modified MC4R that were already described having impaired homodimerization. The first one is the H158R natural variant of MC4R and the second one is a chimera (chim) engineered by replacing the ICL2 of MC4R by the one of the cannabinoid 1 receptor (CB1R)[31]. We confirmed that both SNAP-

tagged constructs were properly targeted to the plasma membrane using a membrane-impermeable SNAP dye (Fig. 7A). Their monomeric nature was also confirmed by molecular brightness assay, although the SNAP-MC4R-H158R displayed a less marked monomeric fingerprint compared to SNAP-MC4Rchim (Fig. 7B). We then tested the function of the MC4R-H158R mutant and the chimera using the Epac-S$^{H187}$ FRET biosensor. The SNAP-MC4R-H158R displayed comparable $E_{max}$ and $EC_{50}$ to the SNAP-MC4R-WT in response to α-MSH (Fig. 8A), as well as a similar behavior as the SNAP-MC4R-WT in response to co-expression of MRAP2 ($1 + 1$), namely a left-shift of the $EC_{50}$ (Fig. 8B). The response of SNAP-MC4Rchim could not be assessed as the maximal response reached the forskolin control and therefore we could not exclude that the assay was saturated. We could however, observe a substantial increase in constitutive activity for the chimera (Fig. 8C).

We conducted parallel experiments using the ebBRET-based miniGs recruitment assay, after verifying that the N-terminal FLAG-tag did not perturb MC4R function (Supplementary Fig. 1C). Consistent with the FRET data, both the FLAG-MC4R-H158R and FLAG-MC4Rchim displayed comparable activation to the FLAG-MC4R-WT (Fig. 8D) in response to MRAP2 co-expression (both $1 + 1$ and $1 + 4$), namely a shift to the left in $EC_{50}$ and a reduction in $E_{max}$ (Supplementary Fig. 11A). Altogether, these results seem to indicate that the observed effects of MRAP2 on MC4R signaling, namely increased potency for Gs and Gq signaling (Figs. 1 and 4), occur primarily due to an interaction between MRAP2 and the MC4R protomer.

Finally, we tested the ability of the MC4R-H158R and the chimera in the presence of the absence of MRAP2 to recruit β-arrestin2 by BRET assays. Interestingly, both the FLAG-MC4R-H158R and the FLAG-MC4Rchim displayed a sizable reduction of β-arrestin2 recruitment ($E_{max}$) to the plasma membrane marker rGFP-CAAX in response to α-MSH (Fig. 8E). The hampered β-arrestin2 recruitment was even more pronounced for the MC4Rchim. For both the mutant and the chimera, co-expression of MRAP2 ($1 + 1$) led to virtually no change in recruitment, whereas co-expression of MRAP2 in a ($1 + 4$) ratio led to a reduction in $E_{max}$ (Supplementary Fig. 11C).

## Putative MC4R–MRAP2 interaction sites

In order to understand how a MC4R monomer/MRAP2 monomer complex might be constituted and influence receptor signaling, we generated a structural homology model of an active MC4R–MRAP2–Gs protein complex with an agonistic ligand based on the MC2R-MRAP1 complex[15]. This is supported by the sequence homology between MRAP2 and MRAP1 (Supplementary Fig. 12 and between MC2R and MC4R (Supplementary Fig. 13). In this homology model, the hetero-dimer interface is predicted to be between the transmembrane helices (TMH) 5 and 6 of MC4R and the membrane-spanning helix of MRAP2. Similarly to the MC2R-MRAP1 complex, additional interactions of the MRAP2 N-terminus with the extracellular loop 3 (ECL3), the TMH7 and the N-terminus of MC4R can be inferred (magenta arrows in Fig. 9).

Of particular note, structural information on the MC2R–MRAP1 complex (PDB ID: 8gy7) does not contain the full-length MRAP1, as the complete intracellular C-terminal tail is missing. This, in turn, precludes any prediction of a potential impact on G protein coupling or homodimerization in the MC4R-MRAP2 homology model. In addition, presumed antiparallel MRAP homodimer formation is not included in the model as it was not observed in the MC2R-MRAP1 structural template.

## Discussion

In this study, we have provided insight into the molecular mechanisms underlying the action of MRAP2 on MC4R cellular function (Fig. 10). Using real-time signaling assays, single molecule microscopy approaches, and molecular modeling we demonstrate that MRAP2: (i) enhances MC4R signaling for at least two different G protein-driven signaling pathways; (ii) decrease the recruitment of β-arrestin; (iii)

modulates MC4R cellular trafficking, (iv) modulates the oligomeric state of MC4R; and (v) enhances Gs-mediated signaling for at least two monomeric forms of MC4R. These findings shed light on how MRAP2 has a direct and modulatory influence on the MC4R, and thus represents a potential important regulator of MC4R-associated functions such as energy consumption and appetite regulation.

The identification of human *MRAP2* mutations found in obese patients highlighted the role of MRAP2 in modulating MC4R signaling[4,25,30]. It was presumed that MRAP2 would function analogously to RAMPs, that can modulate GPCRs by influencing their trafficking to the cell surface, expression, ligand binding, signaling and internalization. Towards this end, an intense effort has been undertaken to understand the role of MRAP2 on these aspects of MC4R physiology and function. In particular, a role of MRAP2 in reducing the capacity of MC4R to form dimers/oligomers was speculated, but until now not experimentally validated[31].

Our findings that MRAP2 increases the capacity of MC4R to activate different G protein subtypes begs the questions (*a*) is this a true reflection of physiology and (*b*) how this might be achieved. In support of the former, there is a precedent for these findings in previous studies that demonstrated a reduction of MC4R signaling in the presence of loss-of-function MRAP2 mutations[4,25,30]. In addition, a more specific increase in Gs and Gq activation by MC4R was observed in HEK293 cells that stably express MRAP2[52]. Interestingly, we could also detect a major increase in signaling with $G\alpha_{15}$, another G protein from the Gq/11 family (Supplementary Fig. 4D). PTX-sensitive activation of endogenous MC4R in GT1-7 murine hypothalamic cell line was previously described, suggesting that Gi activation can be detected in some systems[53], and it would be interesting to see the impact of MRAP2 on Gi pathways.

The mechanisms behind the increase in G protein signaling are still opaque, but can be inferred by some of our additional findings in this study. One possibility is that the increase in the potency to activate G protein would be due to the breakage of the MC4R homo-oligomers, as many dimerization-impaired mutants and engineered chimera were already described to have enhanced G protein signaling[32,33,54,55]. We indeed showed that MRAP2 was able to impact the MC4R monomer-dimer equilibrium, pushing the equilibrium toward a monomeric state (Fig. 7B). Towards this purpose, we tested the naturally occurring homo-dimerization-deficient MC4R-H158H mutation. This is located in the putative MC4R interaction interphase between ICL2 and TM4. We further tested a homo-dimerization-deficient chimeric variant (MC4Rchim) in which parts of MC4R between ICL2 and TM4 were exchanged with CB1R. We observed that both MC4R-H158R and MC4Rchim display Gs signaling comparable to wild type MC4R in absence of MRAP2 and enhanced Gs signaling in the presence of MRAP2 (Fig. 8).

These data suggest that the effects of MRAP2 on the G protein potency would arise from its specific interaction with the protomer rather than on the breakage of the MC4R homo-oligomers. This is also in line with the observation that at a higher ratio, MRAP2 decreases the $E_{max}$ of miniGs recruitment towards the WT (Fig. 2), as well as towards both the MC4R-H158R and MC4Rchim (Fig. 8). This is further supported by the observation that α-MSH binding is enhanced in MRAP2-positive cells, suggesting an increased affinity for the agonist (Fig. 3A, B).

At the same time, the scenario appears to be more complex: MRAP2 decreases the recruitment of β-arrestin2 at the now mono-meric MC4R-WT, and the decrease in β-arrestin recruitment at the MC4R-H158R and the MC4Rchim is aligned to the shift of the oligomeric equilibrium towards the monomeric state. In other words, MC4Rchim has a stronger monomeric fingerprint (Fig. 7) and a worse β-arrestin recruitment (Fig. 8E and Supplementary Fig. 11C) than MC4R-H158R. These results are in line with recent data on the impact of the oligomerization or the oligomer-protomer conformation on β-arrestin engagement of the CXCR4[49] and dopamine 2 receptor[56]. These

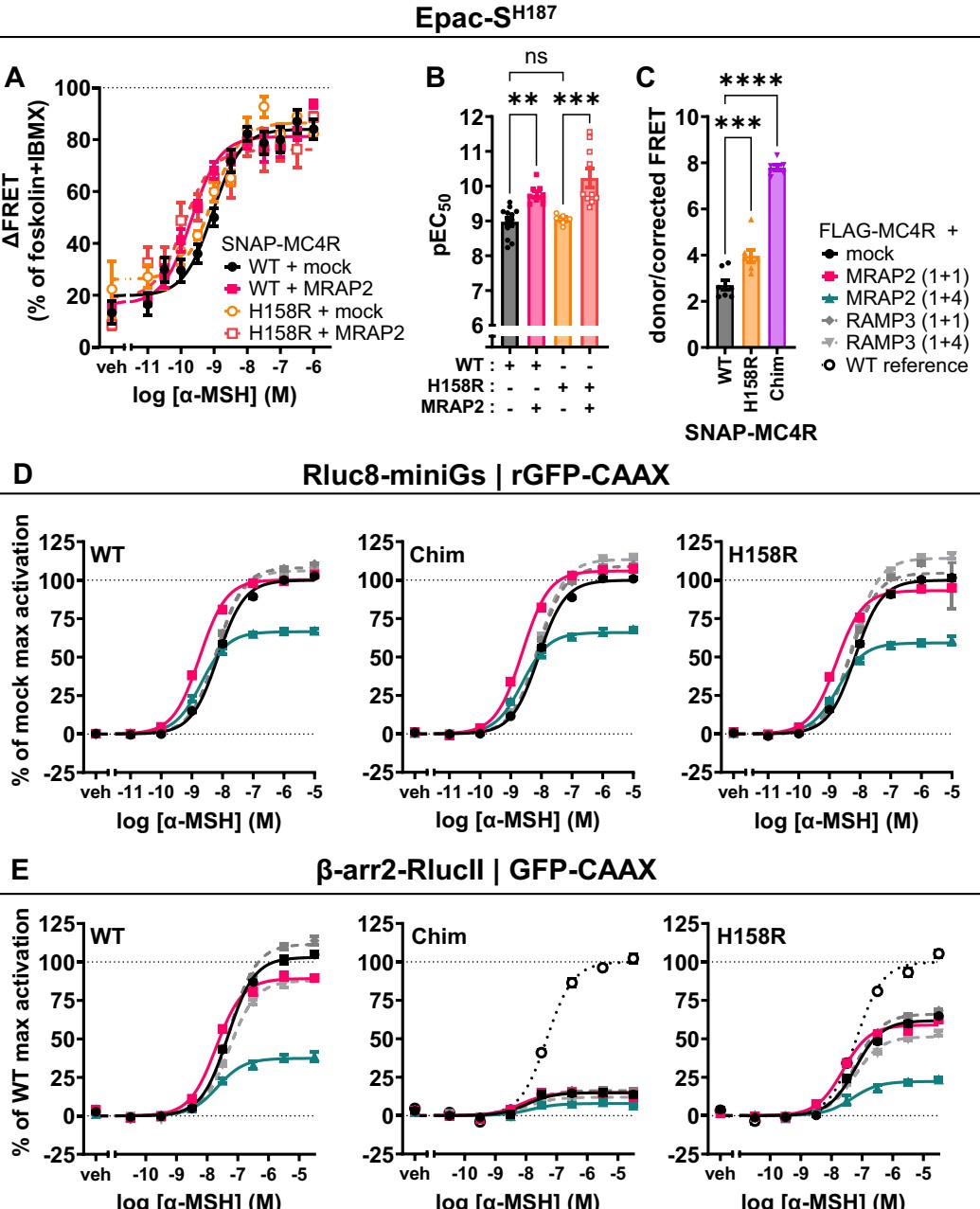

**Fig. 8 | Signaling profile of monomeric MC4R mutant and MC4R chimera are altered by co-expression of MRAP2. A** Average concentration-response curve of cAMP response using α-MSH in transiently transfected HEK293 cells expressing SNAP-MC4R and MC4R-H158R, in the presence or absence of co-transfection with MRAP2 in a 1 + 1 ratio. ΔFRET values are normalized to forskolin response in the presence of phosphodiesterase inhibitor IBMX. Mean ± SEM ($n = 13$) for independent experiments (containing each three technical replicates) are shown. **B** pEC50 values obtained using the Epac-S$^{H187}$ FRET biosensor. Results are represented in a bar chart as mean ± SEM, in which individual points represent pEC50 from independent experiments ($n = 13$) are shown as overlaid dot plot. One-way ANOVA has been performed with Sidak's multiple comparisons as post-hoc test (**: adjusted $p$-value = 0.0035, ***: adjusted $p$-value = 0.0002). **C** Constitutive activity of the MC4R-H158R and the MC4Rchim. Results are represented in a bar chart as mean ± SEM, in which individual points represent biological replicates expressed in donor/corrected FRET ratio from independent experiments ($n = 8$) are shown as overlaid dot plot. One-way ANOVA has been performed with Dunnett's multiple comparisons as

post-hoc test (***: adjusted $p$-value = 0.0007, ****: adjusted $p$-value < 0.0001). **D** Average concentration-response curves of Rluc8-miniGs recruitment to the plasma membrane localization sensor rGFP-CAAX upon stimulation of FLAG-hMC4R-WT, FLAG-hMC4R-H158R, or FLAG-MC4Rchim with α-MSH for 45 min, in absence or presence of MRAP2 or RAMP3 in transiently transfected HEK293-SL cells. BRET values are normalized to maximal response obtained with mock condition. Data are expressed as mean ± SEM of four independent experiments. **E** Average concentration-response curves of β-arrestin2 recruitment to the plasma membrane localization sensor rGFP-CAAX upon stimulation of FLAG-hMC4R(WT), FLAG-hMC4R-H158R, or FLAG-MC4Rchim with α-MSH for 2 min, in absence or presence of MRAP2 or RAMP3 in transiently transfected HEK293-SL cells. BRET values are normalized to maximal response obtained with FLAG-hMC4R-WT. MRAP2 or RAMP3 plasmid DNA is co-transfected with the same amount (1 + 1) or fourfold the amount (1 + 4) of the FLAG-hMC4R plasmid DNA in their respective assay. Data are expressed as mean ± SEM of three independent experiments. Parameters graphs are displayed in Supplementary Fig. 11.

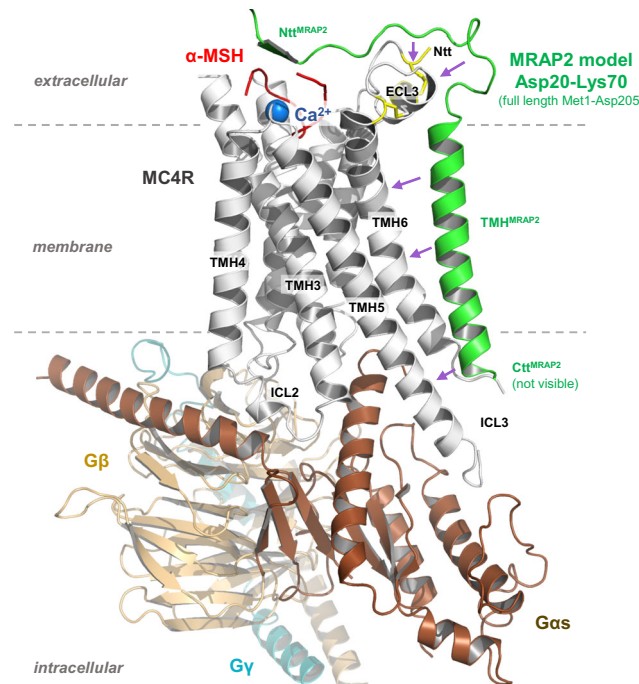

**Fig. 9 | Structural model of an active state MC4R-MRAP2 complex.** In this model, α-MSH (red surface) interacts with the MC4R between the transmembrane helices (TMH) and interconnecting extracellular loops (ECL) as described recently[38,69], together with the essential co-binding factor calcium (blue sphere). The heterotrimeric Gs protein couples intracellularly at the agonist bound MC4R conformation. MRAP2 (green) contacts to MC4R are supposed at TMH5, TMH6, ECL3, and the N-terminus (Ntt) (arrows).

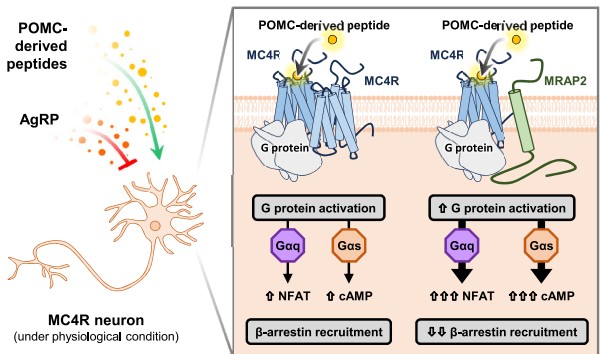

**Fig. 10 | Effects of MRAP2 at the MC4R.** The MC4R is activated by POMC-derived peptides from POMC-neurons the nucleus arcuatus, left side. In addition, AgRP/NPY neurons of the nucleus arcuatus produce AgRP which is in turn axonally transported to the PVN where the function of the MC4R is modulated by AgRP and POMC-derived peptides. Constitutively, the MC4R is in a higher oligomeric state at the plasma membrane of cells (square, left side). As such, it can interact with Gs and Gq as well as β-arrestin2, both constitutively as well as in response to POMC-derived molecules such as α-MSH and result in internalization. Upon co-expression of MRAP2 (square, right side), the equilibrium shifts in favor of monomers. Moreover, the interaction with MRAP2 favors Gs and Gq coupling, whereas ligand-dependent β-arrestin2 recruitment is impaired. These processes are further accompanied by a reduction in constitutive and ligand-dependent internalization.

possibilities of the impact of the oligomerization state shed light on the intricate mechanisms underlying GPCR signaling regulation and warrant further investigation.

The reduction in β-arrestin recruitment could lead to reduced receptor desensitization, which would in turn favor G protein

signaling. However, the presence of MRAP2 induced the same left-shift in the α-MSH-promoted miniGs recruitment in β-arrestin1 and β-arrestin2 double knock-out cells than in SL cells (Supplementary Fig. 7). Those data suggest that β-arrestins play a negligible role in the increase in G protein coupling potency.

As expected with reduced β-arrestin recruitment[29], we also show that MRAP2 co-expression also impairs receptor endocytosis, which would typically result in a build-up of receptors at the cell surface, enhancing G protein signaling responsiveness. We have indeed observed an increase in MC4R cell surface expression upon co-expression of MRAP2 in HEK293, which positively correlates with the amount of MRAP2 detected in the cells by both microscopy and radioligand binding (Fig. 3C and Supplementary Fig. 9). These results are in line with previous observations from radioligand studies on the zebrafish MC4R[27].

This effect is even more striking in N7 cells, an embryonic mouse hypothalamic cell line, in which MC4R is mostly observed intracellularly in absence of MRAP2 (Supplementary Fig. 6). This could either be attributed to a decrease in endocytosis or to an increased in cell surface targeting of MC4R from the secretory pathway, akin to what observed for MRAPs and the MC2R[13]. We shall note here that, with respect to trafficking, MRAP2, as opposed to RAMP3, is not predicted to display a PDZ domain[41].

To understand at the molecular level how MRAP2 could achieve such an impact on MC4R, we developed a homology model between MRAP2 and MC4R. This model indicates potential interaction sites between MRAP2 and MC4R at TMH5, TMH6 and the TMH7-ECL3 transition (Fig. 9). While it is so far unknown how MC4R homodimers are exactly constituted, TMH5 and TMH6 have been identified from various GPCR-GPCR dimer structures as potential interaction sites between the protomers[57], as in the CXCR4 homodimer (PDB ID: 3ODU[58]), or the μOR homodimer (PDB ID: 4DKL[59]). In consequence, binding of MRAP2 at this region should compete with dimerization (or oligomerization) of MC4R.

Moreover, TMH5 and TMH6 of GPCRs are structural rearrangement hot-spots during receptor activation[60], with the highest importance known for TMH6[60]. Our MC4R–MRAP2 model suggests an interaction of both TMH5 and THM6 of MC4R with the transmembrane helix of MRAP2. This interaction could further imply an increased retention of the active state, which might allow for better G-protein coupling observed in our experiment[49]. On the other hand, the TMH5-ICL3-THM6 axis is presumably also important for the spatial adjustment of the arrestin at the receptor. Considering that MRAP2 impairs β-arrestin2 recruitment in our assays, MRAP2 could thus influence the binding of the arrestin as an interfering factor, an aspect that has already been suggested for PKR1[3]. However, as the intracellular loop 3 (ICL3) remains unresolved in our homology model based on the MC2R-MRAP1 structure, it is difficult to predict how this loop would interact with MRAP2.

The increase in potency in activating G proteins could therefore be explained by the increase in affinity for the ligand and/or active state retention by the MRAP2. On the other hand, the decrease in β-arrestin recruitment could be attributed to the breakage of the dimer and/or a steric hindrance from the MRAP2. However, this model does not recapitulate entirely all our observations: the decrease in maximal miniGs recruitment caused by MRAP2 at a ratio of 1 + 4, despite the increased potency, remains unexplained, suggesting the need of further investigations to fully understand the broad and complex repertoire of MC4R regulation by MRAP2.

We shall also remark that to investigate new functional properties of MC4R and MRAP2, the use of fluorescently tagged proteins, heterologously expressed, is the first line of experimentation required to unravel new mechanisms that might be of relevance in physiological conditions. This should be followed to investigation in autologous cell. Unfortunately, in the case of MC4R and MRAP2, this step is hampered by the lack of a human neuronal cell system that recapitulates fully the

paraventricular neurons. We partially addressed this shortcoming by using a mouse hypothalamic neuronal cell line, the N7, that expresses MC4R but not MRAP2. Interestingly, this allowed to show that in the absence of MRAP2, the MC4R is not expressed at the cell surface (Supplementary Fig. 6). In accordance with data obtained in the heterologous system, in N7 cells expressing MRAP2, the enhancement in Gs-mediated cAMP production observed in HEK293 cells is fully recapitulated (left shift of one order of magnitude) (Supplementary Fig. 14).

In summary, our results demonstrate the important role of MRAP2 in modulating MC4R signaling cascades in response to stimulation by the endogenous agonist α-MSH, as well as its synthetic analog NDP-α-MSH, and alterations to the monomer-dimer equilibrium. Albeit monomerization of MC5R by interaction with MRAP1 has been previously shown[61], our observations at MC4R provide evidence directly showing an effect of MRAP2 in regulating an oligomeric GPCR state, which has not been shown experimentally before. While our results showing reduced β-arrestin2 engagement by monomeric mutant MC4R-H158R and MC4R-CB1R chimera are in line with recent publications on the role of oligomerization in this process, more investigation will be needed to make a causal link in the case of MC4R.

The findings reported in our study raise a few interesting aspects relevant to physiology. First and foremost, when translated at the physiological level, our results suggest that MRAP2 expression could play an important role in the modulation of MC4R-related cellular signaling. This is ultimately rooted in the co-expression of MC4R and MRAP2 in the same cells or neurons, as reported in the human brain Atlas (Supplementary Table 1).

Moreover, the recent HYPOMAP dataset confirmed co-expression of MC4R-MRAP2 as well as MC3R-MRAP2[62], demonstrating that distinct cell populations or clusters exist where both proteins are co-expressed. Cell-type specific expression of MRAP2 emerges thus as an additional tool that organisms can use to modulate and bias GPCR signaling in a context-dependent fashion[20].

While until recently the potential effects of MRAP2 on MC4R signaling were considered in the context of Gs/adenylyl cyclase activation[30], our data, together with the results from a recent study published while our manuscript was under revision[63], provide a complementary picture of the broad variety of signaling pathways in which MRAP2 can modulate MC4R function. These pathways include activation of Gq/11, β-arrestin recruitment, cell surface expression, receptor oligomerisation, and internalization. It follows that differences in the MC4R/MRAP2 expression ratios between different tissues/cell types or in pathological conditions could have consequential impacts on the biological actions of MC4R.

Consistent with this notion, in a recent elegant study where MRAP2 was depleted from MC4R neurons[64], the authors reported that MRAP4 KO had an effect on energy metabolism by increasing hyperphagia, impaired glucose homeostasis and insulin sensitivity. In addition, it impacted the sympathetic outflow, which in turn regulates cardiovascular autonomic regulation. This study points to a more complex role of MRAP2 on MC4R physiological functions than previously anticipated; a finding that is consistent with the broad impact of MRAP2 that we observed on the cellular actions of MC4R.

Taken together, our results point to cellular basis that can underlie the role of MRAP2 in the regulation of MC4R-mediated pathways controlling metabolism, satiety and the sympathetic nervous system. Our data pave the way to test this concept for other GPCRs where signal modulation through interaction with either MRAP1 or MRAP2 has been observed[1,65,66].

## Methods
### Cell culture
Different variants of Human Embryonic Kidney cells (HEK293) were employed. HEK293T cell line (ECACC 96121229, Sigma-Aldrich) was used for plater reader and receptor localization and trafficking experiments. HEK293Adherent (HEK293AD) cell line (R70507, Thermo Fisher) was used for molecular brightness experiments (SpIDA). We refer to both clones of HEK293 cells as HEK293 in the main text. Cells were grown in Dulbecco's Modified Eagle's Medium (DMEM, Pan Biotech), supplemented with 2 mM L-glutamine (Pan Biotech), 10% (v:v) heat-inactivated Fetal Calf Serum (Biochrome), 100 µg/mL streptomycin, and 100 U/mL penicillin (Gibco) at 37 °C in a 5% $CO_2$ incubator. HEK293 cells were grown in T75 cm² flasks. Upon reaching approximately 80% confluency, cells were washed with 5 mL phosphate-buffered saline without $Ca^{2+}$ and $Mg^{2+}$ ions (DPBS, Sigma-Aldrich), trypsinised using 3 mL 0.05%/0.02% trypsin/ethylenediaminetetraacetic acid solution (Pan Biotech) and passaged every 2 to 3 days. Cells were routinely tested for mycoplasma infection using MycoAlert Mycoplasma Detection Kit (Lonza).

mHypoE-N7 cells (Cedarlanelabs, accession number CVCL_D462) were maintained in antibiotic-free DMEM/10% FBS/2 mM L Ala-L-Gln at 37 °C/ 5% $CO_2$.

For the BRET-based, sCPF3A-EPAC-venus FRET and radioligand binding assays, HEK293-SL cells[45], a subclone of Ad5-transformed HEK293 cells kindly given by S. Laporte (McGill University, Montreal, Canada), were cultured in DMEM (Wisent Bioproducts) supplemented with 10% newborn calf serum (Wisent Bioproducts). β-arrestin1 and β-arrestin2 double knock-out cells, generated by CRISPR technology from HEK293-SL cells[67], were cultured in DMEM (Wisent Bioproducts) supplemented with 10% fetal bovine serum (Wisent Bioproducts). Both cell lines were cultured with additional 100 UI/mL penicillin and 100 µg/mL streptomycin (Wisent Bioproducts) in an incubator set to 37 °C and 5% $CO_2$. Cells were routinely tested for mycoplasma infection using PCR Mycoplasma Detection Kit (ABM#G238).

### Seeding and transfection
Most transfections for microscopy, FRET-based assays and HTRF IP-ONE assay have been done by using JetPrime (Polyplus), unless specified below. 25-mm glass coverslips and 96-well microtiter plates were coated with poly-D-lysine (PDL) (1 mg/ml) for 30 min, washed twice with PBS, and left to dry before seeding. For microscopy experiments, $3 \times 10^5$ cells were seeded on to clean and PDL-coated glass coverslips (Sigma-Aldrich) into a six-well plate, whereas cell seeding density of 40,000 cells per well was used for plate reader assays. After 24 h, cells were transfected with JetPrime, according to the manufacturer's protocol. For all transfections, MC4R:pcDNA3/MRAP2 ratio was 1 + 4, unless otherwise noted. The empty backbone of pcDNA3 as mock was used throughout to maintain a consistent level of total cDNA. Plasmid DNA constructs for hMC4R, hMC4RChim, and hMRAP2 were subcloned into pcDNA3 and pVITRO2 backbones using Gibson assembly with appropriate SNAP- and eYFP tags, respectively.

For the design of pcDNA-SNAP-hMC4R, the following set of forward and reverse primers were used for vector backbone GCAGATATTAATATCCATCACACTGGCGGCC and TGTCTCCATTCTGCAGAATTCCAGCACACT, respectively. The insert fragments were PCR amplified with the following set of forward and reverse primers, respectively, TCTGCAGAATGGAGACAGACACACTCCTGC and GTGTGATGGATATTAATATCTGCTAGACAAGTCACAAAGGCC.

For the design of pVITRO2-SNAP-hMC4R, the following set of forward and reverse primers were used for vector backbone GTGACTTGTCTAGCAGATATTCTAGAGGGGTAACACTTTGTACTGC and ATTTCGCAGTCTTTGTCCATGGTGGCGATATCACCG, respectively. The hMC4R insert fragments were PCR amplified using these set forward and reverse primers, CAACCGGTGATATCGCCACCATGGACAAAGACTGCGAAATGAAGC and CAAAGTGTTACCCCTCTAGAATATCTGCTAGACAAGTCACAAAGGC.

For the design of pVITRO2-hMRAP2 into pVITRO2 backbone, the following set of forward and reverse primers were used for vector backbone CACACAAAGACCTGGATTGTCATCTAGAGGGGTAACACTTTG

TACTGC and ATTAACCTCTGGGCGGACATGGTGGCGATATCACCG, respectively. The insert fragments were PCR amplified from pcDps-hMRAP2 using these set forward and reverse primers, CAACCGGT-GATATCGCCACCATGTCCGCCCAG and CAAAGTGTTACCCCTCTAG ATCAATCCAGGTCTTTGTGTGAGGT.

For the design of pcDNA-SNAP-hMC4RChim, the following set of forward and reverse primers were used for vector backbone GTGACTTGTCTAGCAGATATTAACTCGAGTCT and CCACGGTGGG TGGAGTTCACGGATCCTGGCGC, respectively. The hMC4RChim insert fragments were PCR amplified using these set forward and reverse primers, GTATAGGCGCGCCAGGATCCGTGAACTCCACCCACC and GTCTCTCTAGACTCGAGTTAATATCTGCTAGACAAGTCA-CAAAGGCCT.

For the design of eYFP-tagged MRAP2, following set of forward and reverse primers were used for the backbone, CACACAAAGACCTGGATT GACTGTGCCTTCTAGTTGCCAG and GAAATTAACCTCTGGGCGGAtc-tagaCTTGTACAGCTCGTCCATG. While the insert MRAP2 was PCR amplified with following forward and reverse primers, respectively, ACGAGCTGTACAAGtctagaTCCGCCCAGAGGTTAATTTCTAAC and CTG GCAACTAGAAGGCACAGTCAATCCAGGTCTTTGTGTGAGGTCT.

**Electroporation of mHypoE-N7 towards confocal microscopy and plate reader assays.** mHypoE-N7, an embryonic mouse hypothalamic cell line were transfected by electroporation using a Neon Transfection System (Invitrogen). One million cells were suspended with 7.5 µg total plasmid DNA, electroporated then seeded across three 25 mm-diameter polylysine-coated 1.5 thickness glass coverslips. SNAP-MC4R was co-transfected at a mass ratio of 1:4 with either eYFP-MRAP2 or pEGFP-N1. SNAP-β1-AR was co-transfected 1 + 4 with pEGFP-N1. For plate reader assays using these cells, SNAP-MC4R was co-transfected at a mass ratio of 1:4:1 with either eYFP-MRAP2 or pcDNA3.1 and EpacS$^{H187}$.

**Transfection via PEI for BRET-based assay and sCFP3A-EPAC-venus FRET experiments.** The day of the transfection, HEK293-SL (WT or β-arrestin double KO) cells are distributed 100 µL per well at a density of 350,000 cells/mL in Cellstar© opaque white 96-well plates (Greiner Bio-One) with 1 µg/mL of total plasmid DNA and 3 µg/mL of 25 kDA linear polyethyleneimine (Alfa Aesar).

**Transfection via FuGENE for radioligand binding assays.** HEK293-SL are seeded into round dishes with a diameter of 150 mm at 7.5 million cells per dish on day one and transfected with 20 µg of DNA and 50 µg of FuGENE® HD transfection reagent (#E2312, Promega) per dish on day two.

**Transfection via Lipofectamine 3000.** HEK293 cells were also transiently transfected following the reverse transfection method using Lipofectamine™ 3000 (Thermo Fisher). Two transfection mixes (A & B) were prepared, in Mix A the DNA was diluted in 25 mL OptiMEM® media, and in Mix B 0.3 mL/well of P3000™ was added. In Mix B, 0.3 mL/well of Lipofectamine™ was added to 25 mL OptiMEM® media and each mix was incubated for 5 min. The DMEM in the HEK293 cells at approximately 70%–85% confluency was removed and washed with DPBS without $Ca^{2+}$ and $Mg^{2+}$, trypsinised and centrifuged as described above. Meanwhile, Mix A containing the DNA was added to Mix B containing Lipofectamine™ mix dropwise and incubated for 15 min at room temperature. The cell pellet was resuspended in 10 mL DMEM-High Glucose media, counted and diluted at a density of 75,000 cells/well. 50 µL of the transfection mix was added into a PDL (Sigma-Aldrich) coated clear white F-bottom 96-well plate, followed by 100 mL of the cell suspension. The plate was incubated at 37 °C in a 5% $CO_2$ incubator for 24 h.

**Epac-S$^{H187}$ FRET-sensor based cAMP assay**
HEK293T cells were seeded into 10 cm Petri dishes ($3.5 \times 10^6$ cells) and after 24 h transfected with the combination of Epac-S$^{H187}$ FRET

biosensor, pcDNA3, SNAP-MC4R (in PCDNA or PVITRO2 including mRuby2 fluorescent reporter) and Epac-S$^{H187}$ biosensor, SNAP-MC4R, and MRAP2 (in PVITRO2 including mRuby2 fluorescent reporter). Plasmid DNA combinations were transfected at a ratio of 5:1:1 or 5:1:4, respectively. Media was changed 4 h after transfection. Twenty-four hours post-transfection, cells were re-seeded into PDL-coated black-wall, black bottom 96-well plates at a density of 40,000 cells per well. 36 h after transfection, cells were washed, and medium was changed to HBSS buffer supplemented with 0.1 % BSA. Plate reader experiments were performed using a Synergy Neo2 plate reader (BioTek) equipped with a monochromator and filter optics. Expression levels of receptor, biosensor, and accessory protein were measured with monochromator optics. For expression of MC4R and MRAP2, cells were excited at 558/20 nm and fluorescence emission was recorded at 605/20 nm while for biosensor the excitation was at 500/20 nm, and emission was at 539/20 nm. For FRET measurements, a range of CFP/YFP monochromators were used, which was excited at 430/20 nm, while 491/30 nm and 541/20 nm were used for emission.

Acceptor baseline fluorescence was measured in 90 µL HBSS buffer. Then, pre-stimulation basal reads were recorded for 5 min. Afterwards, 10 µL of 10× ligand solution in desired dilution series with increasing concentrations was applied to each well and the post-stimulation reads were recorded for a further 25 min. Subsequently, Forskolin/IBMX was added to each well to record maximal stimulation reads for 10 min.

The change in acceptor/donor FRET ratio (ΔFRET) was baseline corrected for each well and timepoint, normalized to 0% baseline and 100% forskolin/IBMX, and plotted in GraphPad Prism 10.5.0. For the concentration-response curves "Dose-response stimulation fit (three parameters)" was applied.

**Nanobody recruitment assay**
HEK293AD cells were seeded in PDL-coated 8-well Ibidi slides at a density of 25,000 cells per well, following with JetPrime transfection the next day, according to the manufacturer's protocol. Plasmid DNA ratios used for co-transfection were as follows: for Nb35/37 recruitment, 1× SNAP-MC4R + 4× pcDNAmock/MRAP2, 1× Nb35-eYFP/Nb37-eYFP, and 3× Gs (tricistronic). Next day, transfected cells were labeled with SNAP-Surface647 dye (NEB) as described earlier. Afterwards, cells were subsequently taken for imaging to an Attofluor Cell Chamber (Fisher Scientific, GmbH) in 1× HBSS supplemented with 0.1 % BSA. A TIRF illuminated Eclipse Ti2 microscope (Nikon), equipped with a 100×, 1.49 NA automated correction collar objective, and with 405, 488, 561, and 647 nm laser diodes coupled via an automated N-Storm module and four iXon Ultra 897 EMCCD cameras (Andor), was used. During imaging, both the cell imaging chamber holding Ibidi slide, and objectives were kept at 37 °C. The automated objective collar was kept on and hardware autofocus was activated.

An excitation wavelength of 514 nm to quantify Nb35 recruitment via eYFP and an excitation of 633 nm to quantify receptor on the membrane via SNAP-Surface647 were used. Moreover, 405 nm wavelength was used for the excitation of CFP-tagged Gs.

The acquisition of movies was performed at 4 s per frame for 400 frames. An initial baseline measurement of 25 frames was before NDP-α-MSH was added at a concentration of 1 µM to the well. Data were measured across single cells (technical replicates, generated across multiple transfections). Fluorescence intensity values were extracted from ROIs as single cells by employing ImageJ. Background correction and normalization was made for pre-stimulation and post-stimulation frames.

**IP$_1$ accumulation assay**
Cellular IP1 as a readout for Gq signaling was measured using a commercial HTRF competitive immunoassay (IP-One Gq kit, Part #62IPA-PEB, Revvity) as described in the manufacturer's protocol. For this

 

purpose, HEK293T cells were seeded into 10 cm Petri dishes ($3.5 \times 10^6$ cells) and after 24 h transfected with the combination of plasmids (pcDNA3+SNAP-MC4R) and (SNAP-MC4R + MRAP2) at a ratio of 1:4, using JetPrime according to the manufacturer's protocol. Media was changed 4 h after transfection. Twenty-four hours post-transfection, 20 μL of cell suspension at a density of 15,000 cells per well was re-seeded into opaque white flat-bottom 384-well plates (Corning) and left to equilibrate for 2 h at 37 °C in a 5% CO2 incubator. Afterwards, the cells were stimulated with ligand concentrations (dilution series for concentration-response curve) prepared in stimulation buffer (HBSS supplemented with 20 mM LiCl) for 2 h under cell culture conditions. Following incubation, the FRET acceptor ($IP_1$-d2) and the FRET donor (anti-$IP_1$-Cryptate antibody) were reconstituted and diluted in lysis and detection buffer from 6× to 1× working stocks, as described in the manufacturer's protocol. 3 μL each of both solutions was added per well separately in the order described, and the plate was incubated on a shaker for an hour at room temperature. Fluorescence emission was detected at 620 nm and 665 nm at the plate reader (Synergy Neo2 plate reader, BioTek). The HTRF ratio was calculated by dividing the detected emission at 665 nm by the detected emission at 620 nm. The $EC_{50}$ values were calculated using the GraphPad Prism 10 software (GraphPad Software, San Diego, CA, USA).

## NFAT reporter assay
HEK293 cells were transfected 24 h after seeding. For determination of PLC activation (read-out for Gq/11 signaling) via nuclear factor of activated T-cells (NFAT) reporter gene assay, cells were transfected with 45 ng plasmid-DNA, 0.45 μl Metafectene (Biontex, Munich, Germany) and with 45 ng NFAT reporter DNA per well in MEM without supplements.

For PLC activation, the NFAT responsive element (NFAT-luc, pGL4.33) was co-transfected with the plasmids for MC4R and MRAP2 in a ratio of 1 + 1 (1:1) or 1 + 4 (1:4). 48 h post-transfection, cells were challenged with α-MSH (10 mM to 1 nM) in MEM without supplements for 4 h at 37 °C and 5% CO₂. The stimulation was stopped by discarding the media and cell lysis was induced by the addition of 1× passive lysis buffer (Promega) and horizontal shaking for 15 min at room temperature.

For measurement of luciferase activity, 10 μl lysate were transferred into a white opaque 96-well plate. Injection of 40 μl firefly luciferase substrate (Promega) and measurement of luminescence was performed with a plate reader (Viktor Nivo, Perkin Elmer).

## Synthesis of AlexaFluor647-NDP-α-MSH
NDP-α-MSH was synthesized by solid phase peptide synthesis based on the 9-fluorenylmethyloxycarbonyl (Fmoc)/tert-butyl (tBu) strategy, carried out at 15 μmol scale. An eightfold molar excess of N-α-Fmoc-protected amino acid, Oxyma, and diisopropylcarbodiimide in DMF was used for automated robot synthesis performed on a SYRO I peptide synthesizer (Witten, Germany). Coupling reactions were carried twice with a reaction time of 40 min each. For Fmoc cleavage, a solution of 40 % (v/v in DMF) piperidine was applied for 3 min and 20 % (v/v in DMF) piperidine for 10 min. Alexa647 was coupled at the N-terminus of NDP-α-MSH, still bound to the resin. The fluorescent peptide was cleaved from the resin with trifluoroacetic acid and purified by high-pressure liquid chromatography to >95% homogeneity. Identity was confirmed by electrospray ionization mass spectrometry (HCT, Bruker).

## Time-lapse imaging with confocal microscopy
HEK293T cells ($3 \times 10^5$ cell per well) were seeded on 25 mm glass coverslips in a 6-well plate, which were pre-coated with PDL. After 24 h, cells were transfected with 2.5 μg DNA in total of SNAP-MC4R and MRAP2-eYFP constructs (1:4 ratio) using JetPrime according to the manufacturer's protocol. Media was exchanged to fresh media 4 h after

transfection. Next day, transfected cells were labeled with SNAP-Surface Alexa Fluor 647 dye. Afterwards, coverslip with labeled cells was placed in the Attofluor cell chamber (Thermo Fisher Scientific) with 450 μL of the imaging buffer (HBSS supplemented with 0.1% BSA) and the chamber was placed in the placeholder for image acquisition. Pre-stimulation images were taken for the first 3–5 min and then 50 μL of agonist NDP-α-MSH was carefully added on top of cells to the final concentration of 1 μM in the chamber and image acquisition was performed for 30 min post-stimulation.

Imaging for Figs. 5 and 7 was performed using Leica SP8 confocal microscope system with 40×/1.25 NA oil immersion objective, a photon counting hybrid detectors and a white light laser (WLL). 488 and 633 nm lines of the WLL were used for excitation and emission bands of 520–600 nm and 650–730 nm were used for detection of eYFP and SNAP-Surface Alexa 647, respectively. Imaging format was xyt, and image size was set to 1024 × 1024 pixels and 100 Hz scan speed. Image analysis was performed using ImageJ.

For mHypoE-N7 and HEK293 used in Fig. 3 and Supplementary Fig. 6 multitrack confocal z stacks were acquired on a Zeiss LSM800 Airyscan microscope (Zen software) using a 63× 1.4 NA objective at 50 nm lateral pixel resolution. Laser excitation/emission collection (nm) was 488/500–550 for EGFP/eYFP, 561/580–630 for SNAP surface 549 and 640/650-700 for AF647. Maximum intensity projections were constructed in Fiji and all images are presented on a common intensity scale.

Prior to microscopy imaging experiments coverslips with cells (either HEK293 or mHypoE-N7) expressing SNAP-MC4R were with 1 μM SNAP-Surface647 dye or SNAP-Surface549 in complete DMEM (Pan Biotech) for 25 min and kept in the incubator at 37 °C and 5% CO₂. Excessive dye was washed by exchanging with fresh medium with an interval of 5 min each for three times.

## Fluorescence cross-correlation spectroscopy assays
Transfected cells (same conditions as in Time-lapse imaging with confocal microscopy) were plated onto poly-L-lysine-coated glass coverslips 12 h prior to imaging to ensure proper adhesion. Fluorescence line-scan measurements were performed using a Leica SP8 laser scanning confocal microscope, operating at a scanning frequency of 1800 Hz. The spatial resolution was 50 nm per pixel with 256 pixels per line. A typical acquisition consisted of 500,000 lines. Excitation wavelength was set at 514 nm (ArIon laser) to excite e-MRAP2, 561 nm (solid state diode) to excite SNAP-Surface549, and 633 nm (HeNe laser) to excite NDP-α-MSH-Alexa647. Linescan analysis to generate Spatial-Temporal Correlation Spectroscopy functions was conducted as previously described[68].

Briefly, post-acquisition, line-scan images were subjected to manual inspection to identify and exclude regions exhibiting significant temporal intensity drop due to photobleaching. Additionally, spatial segments displaying abrupt intensity changes were excluded to avoid artifacts in the downstream analysis. The images were then detrended by equalizing the mean along the time axis for a given bin size (typically 10,000 lines) though the addition of unit numbers at random locations. 2D correlation was carried out on the detrended image to obtain a STICS function, which was logarithmically binned along the time dimension. This analysis was conducted using custom written routines in Python (see "Code Availability") and displayed using IgorPro 9.05 (Wavemetrics). The resulting STICS profiles exhibit a plume-like shape, broadening spatially and decaying temporally, consistent with diffusion-driven transport. Cross-sectional profiles along spatial coordinate '0' were fitted to a 2D autocorrelation function[68], where the decay time relates to the diffusion coefficient.

## NanoBiT arrestin recruitment assay
HEK293 cells (75,000 cells/well) were seeded in white bottom 96-well plate and reverse co-transfected with Lipofectamine 3000, whereby

the plasmid encoding the MC4R-LgBiT (50 ng/well) and the SmBiT-β-arrestin 2 (10 ng/well) were used. 24 h post-transfection, the cells were rinsed once with assay buffer (1× HBSS, 24 mM HEPES, 0.1% (w:v) BSA, 3.96 mM NaHCO$_3$, 1 mM MgSO$_4$, 1.3 mM CaCl$_2$ (2H$_2$O), pH 7.4) and the plates were pre-equilibrated in the dark for 1 h at 37 °C with 90 μL of assay buffer. Post-equilibration, 25 μL/well of a 5× solution of the Nano-Glo® Live cell reagent (NanoLight Technology) was added, and luminescence readings were taken every minute at 37 °C until the signal was stable (3–5 min), representing the basal activity. Immediately after, 10 μL of agonist/vehicle were added and luminescence was further recorded for 60 min (no lens, 0.5 s integration time and 1 min intervals at 37 °C using a CLARIOstar® Plus Multimode Plate Reader (BMG Labtech, Germany). The final volume in each well was 125 μL. To account for differences in expression/cell density, the average of at least 3 stable pre-readings was used to normalize each well response before proceeding with further analysis using the area under the curve parameter.

## Molecular brightness analysis with SpIDA

HEK293AD cells (3 × 10$^5$ cells/well) were seeded on clean glass coverslips in six-well plates and were transfected with MC4R and MRAP2 constructs for overnight using JetPrime as a transfection reagent according to the manufacturer's protocol. For SNAP-tag incorporated receptors, labeling was performed as described above using SNAP-Surface647 dye. Coverslips with transfected cells were placed in an Attofluor cell chamber (Thermo Fisher Scientific) and supplemented with Flourobrite DMEM supplemented with 2 mM L-glutamine (Pan Biotech), 10% fetal calf serum (Biochrome), 100 μg/mL streptomycin, and 100 U/mL penicillin (Gibco).

SpIDA imaging was performed using a commercial laser-scanning confocal microscope (Leica SP8) equipped with a 40×/1.25 NA oil immersion objective, a WLL, and photon counting hybrid detectors; 514 and 633 nm lines of the WLL were used for excitation, and emission bands of 520 nm to 600 nm and 650 nm to 730 nm were used for detection of eYFP and SNAP-Surface647, respectively. Imaging format was xy, and image size was set to 512 × 512 pixels with 50 nm pixel size and 100 Hz scan speed. Image analysis was performed using ImageJ. Polygonal region of interest (ROI) selection was implemented and areas with inhomogeneous fluorescence distribution were avoided.

## Homology modeling of MC4R–MRAP2 complex

A structural homology model of an agonist–MC4R–MRAP2–Gs complex was designed based on the recently determined ACTH–MC2R–MRAP1–Gs complex (PDB ID: 8gy7)[15] and the NDP-α-MSH–MC4R–Gs protein complex (PDB ID: 7piv)[38] as structural templates. Both MC2R-MC4R as well as MRAP1-MRAP2 share high sequence similarities of ~60% and, respectively. Full sequences of MRAP1 and MRAP2 were manually aligned (Supplementary Fig. 12). This high similarity supports the idea of potential homology between the two receptor-MRAP complexes, however, differences in detail must be considered especially with respect to ligand binding modes. The complexes were superimposed at the receptor's transmembrane region, and MRAP was merged from the ACTH–MC2R–MRAP1–Gs complex into the high resolution (2.58 Å) NDP-α-MSH–MC4R–Gs protein complex. NDP-α-MSH was substituted by α-MSH as known from the α-MSH–MC4R complex[69], with a lower resolution of ~3 Å. The sequence of MRAP1 was substituted by corresponding amino acid residues of MRAP2. Of note, the used alignment between MRAP1 and MRAP2 sequences is without gaps (Suppl. Fig. 9) and is, therefore, different to a recently supposed[15]. The N-terminal MRAP1 '$_{18}$LDYL$_{21}$' motif is suggested to be an MRAP1 specific feature compared to the MRAP2 sequence, and is significant to mediate the effect of increased ACTH action at MC2R. This might be different to (structurally unknown) MC4R-MRAP2-ACTH complexes, whereby it was already shown that co-expressed MRAP2[70,71] increases ACTH binding and

signaling to a high extent, in contrast to α-MSH. In MRAP2 the exact MRAP1 "$_{18}$LDYL$_{21}$" motif is missing or is potentially structurally substituted by a "$_{25}$YEYY$_{28}$" motif with similarities as considered in the alignment presented in Supplementary Fig. 13. For all steps of structural modifications, the software SYBYL-X 2.0 (Certara, NJ, US) was used and the initial complex model was optimized by energy minimization with the Amber99 force field until converging at a termination gradient of 0.05 kcal/(mol × Å) under constrained backbone atoms. Side chain orientations were optimized by a 2 ns molecular dynamics simulation (MD) with constraint backbone atoms. Finally, the entire complex model α-MSH–MC4R–MRAP2–Gs was energetically minimized without any constraint.

## BRET-based assays

Plasmids encoding for Rluc8-C1_Rluc8-miniGs[42], pcDNA3.1hygro(+)_p63RhoGEF-RlucII, pcDNA3.1hygro(+)_Rap1GAP-RlucII[42], pcDNA3.1-hygro(+)_β-arrestin2-RlucII, pcDNA3.1(+)_rGFP-CAAX, pcDNA3.1(+)_rGFP-FYVE[45], pcDNA3.1(+) MRAP2[72], and β-arrestin1 and β-arrestin2 were already described. RAMP1 and RAMP3 were kindly provided by S. M. Foord (GlaxoSmithKline) and subcloned into pcDNA3 as previously described[73]. pcDNA3.1_Gα$_q$ and pcDNA3.1_Gα$_{15}$ were purchased from cDNA Resource Center. New pcDNA.3.1(+)_MC4R-RlucII was generated from 3 × HA-hMC4R-RlucII using Gibson assembly to remove the N-terminal 3 × HA tag, and to add a full Kozak consensus sequence before the receptor, to revert the silent mutation c.G993A to refseq sequence, and to replace the linker with GGGGSKLPAT between the human MC4R (MCR040TN00, cDNA Resource Center) and the RlucII[45]. For the BRET assays, either SNAP-MC4R or FLAG-MC4R were used respectively for MC4R-WT evaluation or MC4R-H158R and MC4Rchim evaluation. The SNAP-MC4R is in a pVITRO2 dual expression vector (Invivogen), in which mRuby2 is expressed under the FerL promoter, and the SNAP-MC4R is expressed under the FerH promoter. The SNAP-MC4R is N-terminally tagged with consecutively an Ig kappa ladder signal peptide (Igκ), a hemagglutinin A (HA) tag, a Strep-Tag®II (ST) tag and a SNAP-tag® (Igκ-HA-ST-SNAP-MC4R). The FLAG-MC4R is in a pcDNA3.1(+) expression vector (Invitrogen) and was previously described[74]. Cloning for H158R was performed using Q5 mutagenesis with forward primer TCTCCAGTACcgcAACATTATGAC and reverse primer GCATAGAAGATAGTAAAGTAC on pcDNA3.1(+)_FLAG-hMC4R(WT) template. Cloning for MC4RChim was performed using forward primer agctggctagcgtttaaacttaagcttgccaccATGGATTACAAG GATGACGACGATAAGGggtaccATGGTGAACTCCACCCACC and reverse primer tcagcgggtttaaacgggccctctagaggctcaTTAATATCTGCTAGACAA GTCACAAAGGCC on pcDNA3_SNAP-MC4R-Chim7 template previously described.

Total plasmid transfected was adjusted to 1000 ng with pcDNA3.1(+) and fixed plasmid amounts were as follows: 25 ng MC4R-RlucII, 400 ng rGFP-FYVE, 300 ng rGFP-CAAX, 5 ng p63RhoGEF-RlucII with 100 ng Gα$_q$, 10 ng p63RhoGEF with 50 ng Gα$_{15}$, 5 ng β-arrestin2-RlucII, 5 ng Rluc8-miniGs. For β-arrestins rescue experiments, both β-arrestins were reintroduced at 100 ng each. MC4R amounts were adjusted to be around the middle of the linear range of each functional assays, which is 100 ng for miniGs recruitment assay, 140 ng for β-arrestin recruitment assay, 400 ng for p63RhoGEF recruitment assay with Gα$_q$, 10 ng for p63RhoGEF recruitment assay with Gα$_{15}$. MRAP2 amounts were used at varying amounts, but RAMP3 amounts were only tested at the higher of the MC4R:MRAP2 ratio used for some assays.

Two days after transfection, plates were washed with PBS supplemented with 1 mM CaCl$_2$ and 1 mM MgCl$_2$ (PBS + CM), and equilibrated at 37 °C for approximately 1 h in Tyrode HEPES buffer (NaCl 137 mM, MgCl$_2$ 1 mM, NaHCO$_3$ 11.9 mM, NaH$_2$PO$_4$ 3.6 mM, KCl 0.9 mM, glucose 5.5 mM, CaCl$_2$ 1 mM, with pH adjusted to 7.4 with NaOH). Total fluorescence of the rGFP (λ$_{excitation}$ 485 nm, λ$_{emission}$ 538 nm, auto cutoff 530 nm) and/or the mRuby2 (λ$_{excitation}$ 550 nm, λ$_{emission}$ 615 nm, cutoff set to 610 nm) was red before BRET experiments using FLEX

station II (Molecular Devices), respectively to control for plasmid competition of the biosensors and dual expression of the pVITRO2_SNAP-MC4R.

For dose-response curve experiments, cells were incubated with increasing concentrations of α-MSH (GenScript) with 0.01% m/V bovine serum albumin (Biobasic) for 45 min for G protein activation assays, for 2 or 3 min for β-arrestin2 recruitment assay and for 60 min for internalization assay. Prolume Purple methoxy e-Coelenterazine (NanoLight Technology) was added 10 min before reading (1 µM final). Light measurements at 410/80 nm (donor) and 515/20 nm (acceptor) channels were integrated for 0.1 s at 37 °C on Tristar$^2$ LB942 (Berthold Technology). BRET is calculated as the ratio of the light emitted in the acceptor channel over the light emitted in the donor channel.

Results were normalized to the α-MSH-induced change in BRET ratio of the mock condition for each individual experiment.

$$\alpha\text{-MSH-induced change in BRET}(\%)$$
$$= \frac{\text{BRET}_{condition} - \text{bottom asymptote}_{condition}}{\text{span}_{mock}} *100$$

Concentration-response curves and parameters, such as bottom asymptote, span and $\log EC_{50}$ were calculated by GraphPad Prism using 3-parameters non-linear regression. $E_{max}$ is calculated manually as span ratio over the mock. It is to note that all calculations excluded changes in basal.

$$E_{max}(\%) = \frac{\text{span}_{condition}}{\text{span}_{mock}} *100$$

Statistical analysis were performed on GraphPad Prism (10.3.1).

### sCFP3A-EPAC-Venus cAMP FRET assay

HEK293-SL cells were cultured and transfected like for the BRET experiments with 1000 ng of pcDNA3.1_sCFP3A-EPAC1(δDET + T781A + F782A)-venus and increasing amount of either pcDNA3.1(+)_MC4R-WT, pcDNA3.1(+)_FLAG-MC4R-WT or pcDNA3.1(+)_MC4R-WT-RlucII up to 200 ng, total DNA being maintained to 1200 ng with pcDNA3.1(+) empty vector.

Method to generate pcDNA3.1_sCFP3A-EPAC1(δDET + T781A + F782A)-venus was previously described[75].

Two days after transfection, cells were washed with PBS + CM and equilibrated for 30 min in THB. Cells were incubated with increasing concentrations of α-MSH (GenScript) with 0.01% m/V bovine serum albumin (Biobasic) for 25 min. Fluorescence signals at 485/20 nm and 535/25 nm were read on Spark microplate reader (Tecan) with excitation at 430/20 nm. FRET values were calculated as light emitted in the 485 nm channel divided by sum of the light emitted in both 485 nm and 535 nm channels. Baseline-corrected values (representing the α-MSH-induced change in FRET) were calculated by subtracting the bottom asymptote calculated by GraphPadPrism (10.3.1) for each independent experiment.

### Membrane preparation for radioligand binding experiments

On day one, HEK293-SL cells grown in DMEM supplemented with 10% NCS and 1% PenStrep are seeded into round dishes with a diameter of 150 mm at 7.5 million cells per dish.

On day two, cells are transfected with 20 µg of DNA and 50 µg of FuGENE® HD transfection reagent (#E2312, Promega) per dish as recommended by the manufacturer protocol. Plasmid construction for MC4R with no N-terminal tag and a stronger Kozak sequence was generated by PCR amplification using forward primer agcgtttaaacttaagcttggtaccgccaccATGGTGAACTCCACCCACCGTGGG and reverse primer tcagcgggtttaaacgggccctctagaggctcaTTAATATCTGCTAGACAAGTCACAAAGGCC. Insert was subcloned into pcDNA3.1(+) by Gibson assembly using KpnI and XbaI restriction enzymes (plasmid name

pcDNA3.1(+)_kozak_MC4R-WT to differentiate from the other pcDNA3.1(+)_MC4R-WT) used for the FRET_EPAC experiments). The MC4R sequence is confirmed to be the same as for FLAG-MC4R-WT. DNA amounts were 10 µg pcDNA3.1(+)_kozak_MC4R-WT, 10 µg of pcDNA3.1(+) empty vector, 10 µg of pcDNA3.1(+)_MRAP2 or 10 µg of pcDNA3_RAMP3.

On day three, the media is changed.

On day four, cell membranes are isolated. Briefly, cells are washed with PBS at room temperature. Warmed TripLE™ Express (Gibco # 12604-013) is added for 2 to 5 min at 37 °C. Cells are collected with growth medium on ice. Cells are centrifuged for 5 min at $500 \times g$ at 4 °C. Cells are washed with PBS, then centrifuged again for 5 min at $500 \times g$ at 4 °C. Supernatant is discarded and the pellet is weighed. Cells are resuspended in 10 mL of ice-cold membrane preparation buffer (HEPES 50 nM, Halt™ protease inhibitor cocktail (#78429, Thermo Scientific) 1×, EDTA 2 mM, and MgCl2 10 mM). Suspension is homogenized with an Ultra-Turrax T25 homogenizer (Janke & Kunkel IKA-Labortechnik) at maximum speed three times for 7 s each time (rest on ice between cycles). Suspension is transferred in tubes suitable for a high-speed centrifuge and centrifuged at $50,000 \times g$ for 30 min (Avanti JXN-26 with JA-25.50 rotor). Supernatant is discarded and tubes are inverted to drain liquid for 1–2 min. Pellet is resuspended in 1 mL of ice-cold membrane preparation buffer per gram of pellet. Suspension is homogenized again at maximum speed for 3 times for 7 s each time (rest on ice between cycles). Suspension is passed through a 1 mL syringe with a 23G needle. Protein content is quantified by Pierce™ BCA Protein kit as (#23225, Thermo Scientific). Suspension can be diluted, aliquoted, and stored at −80 °C.

### Scintillation proximity assay (SPA) for radioligand binding experiments

Membranes from HEK293-SL previously prepared (in a 0.13 mg/mL suspension) and Wheatgerm Agglutinin Coated PVT SPA beads (SPA beads; RPNQ0001, Revvity) (in a 2 mg/mL suspension) are pre-incubated together (1:1 volume: volume) in the binding buffer (HEPES 20 mM adjusted to pH 7.4 with 1 KOH, 0.1 mM $MnCl_2$, 1 mM $CaCl_2$, 0.5% (w/v) BSA, 1 µg/ml protease inhibitor cocktail) for 60 min on a Nutator at 4 °C in the dark. Suspension is centrifuged at $1000 \times g$ for 10 min at 4 °C and supernatant is discarded to remove unbound membrane. Pellet is resuspended with the same volume of binding buffer. 1.5 µL of Setmelanotide (HY-19870/CS-6399, MedChem Express) prepared in DMSO is added into the non-specific binding wells for a final concentration of 5 µM, and 1.5 µL of DMSO is added in the test wells (total binding). 100 µL of the SPA beads and membranes suspension are added into wells, and plate is incubated for 2 h in dark with a low shaking (-150 rpm). The plate is sealed with TopSeal™-A Plus (6050185, Revvity), and an underpad is added to avoid luminescence. 50 µL of $I^{125}$-NDP-α-MSH (Revvity, #NEX352) are added into the wells (seal is open and then closed). The plate is then incubated as before for at least 20 h before integrating the SPA signal on MicroBeta2™ microplate counter (model 2450, PerkinElmer) for one minute.

Nominal concentrations of radioactive ligand are calculated post-experiment using decay catastrophe calculations for $I^{125}$. $K_D$ and $B_{max}$ are extrapolated using the "One site–Total and nonspecific binding" equation from GraphPad Prism. $B_{max}$ of the +MRAP2 and +RAMP3 conditions are normalized to the mock condition. Statistical significance of the effects of MRAP2 and RAMP3 on the affinity as $pK_D$ and the expression of MC4R as normalized $B_{max}$ was assessed using ordinary one-way ANOVA.

### Statistics & reproducibility

No statistical method was used to predetermine sample size. Radioligand binding experiments were repeated independently five times, but two datasets were excluded as the negative controls were not correct. The experiments were not randomized. The investigators

were not blinded to allocation during experiments and outcome assessment. Details of statistical analyses, including the specific tests used, are provided in the figure legends and relevant sections of the "Methods". All experiments reported in the main Figures were repeated independently in at least three biological replicates with similar results.

## Reporting summary

Further information on research design is available in the Nature Portfolio Reporting Summary linked to this article.

## Data availability

The source data for the images, charts and graphs of Figs. 1–8 are provided as Source Data files. The raw microscopy data underpinning the charts and graphs reported in this manuscript are available under restricted access due to their very large size, and can be obtained by request to the corresponding author. Please allow for two weeks to address the request, and data will be shared via an appropriate online repository for the duration of a week. Source data are provided with this paper.

## Code availability

A Python version of the code used to generate the STICS functions can be found on GitHub https://github.com/LynnLangstrumpf/Master-Thesis.

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

## Acknowledgements

We thank Sabine Jyrch for excellent technical assistance. We would like to thank Alexei Sirbu for his helpful support. This work was supported by the Deutsche Forschungsgemeinschaft (DFG) (German Research Foundation) through SFB1423, Project-ID 421152132, subprojects C03 (to P.A., M.J.L.), B02 (to H.B.), A01/Z03 (to P.S.), A04/B01/Z03 to (A.G. B.-S.). P.A.

would like to acknowledge generous funding from the Leverhulme Trust RL-2022-015. Funding was received through Germany's Excellence Strategy–EXC 2008–390540038–UniSysCat (Research Unit E) (to G.K. and P.S.) and by the European Union's Horizon 2020 Research and Innovation Programme under the Marie Skłodowska-Curie grant agreement No 956314 [ALLODD] (to P.S.). UKRI funding to P. McC. CIHR funding (PJT-183758) to M.B. FRQS PhD scholarship to S.-A.L. We would like to thank the Advanced Light Microscopy (ALM) facility at MDC, Buch, Germany.

## Author contributions

H.B., P.A., P.McC., and M.B. designed research. I.S., S-A.L, G.K., V.C., A.M., A.B., Z.C.U.K., M-J.B, J.T., P.S., A.G.B-S. performed research. I.S., S.-A.L., V.C., P.A., A.B., A.M., Z.C.U.K., H.B., M.B. data analysis. G.K. and P.S. performed, analyzed, and visualized structure and sequence analysis. I.S., S-A.L., V.C., P.A., H.B., graphical analysis. P.A. H.B., P.McC., M.B., I.S., S-A.L., M.J.L., G.K., and P.S. wrote and revised the manuscript. H.B., P.A., M.B., P.McC. supervision of experiments. H.B., P.A., A.G.B-S., M.J.L., P.S., P.McC.M.B. funding acquisition

## Competing interests

The authors declare no competing interests.
