## [Transparent Peer Review file · Nature Communications]

MRAP2 modifies the signaling and oligomerization state of the melanocortin-4 receptor

Corresponding Author: Dr Paolo Annibale

Version 1:

Reviewer comments:

Reviewer #1

(Remarks to the Author)

This is an interesting study looking at the role of MRAP2 on MC4 receptor pharmacology. The role of accessory proteins and their impact on GPCR signal transduction is a very important and is gaining more traction in the area.

Overall I find most of the work here has been performed in a diligent and well contracted manner. There are some areas however that I think need improvement.

1. I would like to see the impact of the MRAP2 interaction MC4 receptor agonist binding. Are the affinities changed for the agonists if the MRAP2 is present? I think this is critical for the story here as it will impact downstream signalling.
2. While there has been effort to use to ratio's of MRAP2 to MC4 receptor expression, I would like to see a situation assay confirming that the amounts used are appropriate. This is critical since how can the investigators be confident that ALL cells express both proteins.
3. To this end, have the investigators consider he use of some for of molecular glue to ensure that the MRAP2 and MC4R are "forced" together. While this might impact signalling it will give confidence in the interaction.
4. RAMP proteins have been shown to impact signalling (e.g. CLR, GCGR, AMYR) through the C-terminal tails. Did the investigators consider remove the MRAP2 tail to see if effects are removed.
5. RAMP3 has a PDZ domain linked to trafficking. Does MRAP2 have the same effect?
6. Have the investigators considered the use of BRET to show MRAP2 and MC4R interactions - both on the cell surface and intercellular?
7. I find the homology section speculative. I think this section would benefit from MD simulations which are very intensive. If the investigators which to keep this section in the manuscript, I think it would be better in the supporting information as without MD simulation I am not sure it is adding much to the study.

Reviewer #2

(Remarks to the Author)

This manuscript from Sohail, Laurin, Bouvier, McCormick, Annibale and Biebermann demonstrates that the single-pass transmembrane protein, MRAP2, acts as an accessory protein (similar to a RAMP) for MC4R to modulate the plasma membrane localisation and oligomerisation state of the receptor. The team use an array of FRET, nanobody and BRET biosensors to show increased potency and/or efficacy of alpha-MSH signalling (cAMP) via the MC4R with MRAP2 co-transfection. MRAP2 co-transfection also increased the proportion of MC4R at the plasma membrane, and decreased both the recruitment of beta-arrestins to MC4R and the relocation of MC4R to endosomes. TIRF imaging and calculation of molecular brightness suggests that MRAP2 favours monomeric MC4R. Consistent with this, MRAP2 was still able to modulate G protein coupling of MC4R constructs previously established to have impaired homodimerization, but these receptors showed impaired beta-arrestin recruitment compared to wild-type.

The team have generated an impressive amount of data using a wide variety of experimental end-points. The figures are clear and well-organised.

Below are some comments for the authors to address.

1. We think it is important to confirm that the amount of MC4R protein in the transfected cells is consistent across the different

transfection conditions (i.e. does switching from co-transfection with empty vector, to 1+1 MRAP2 or 1+4 MRAP2 impact the amount of MC4R produced?). A change in the amount of MC4R expressed in the cells could also impact EC50 and/or EMax values.

Similarly, please confirm that transfecting increasing amounts of MRAP2 results in increased expression of MRAP2 protein (this is important to confirm the proposed saturable effect on EC50 suggested for panel 1B).

2. Part of the rationalisation for the current study was to more precisely characterise the MRAP2/MC4R interaction i.e. Results pg 5, line 118 "... wide range of transfection ratios... highly variable outcomes are reported, without proper explanation for their selection"

Given this statement, it seems important to clearly rationalise why different transfection ratios are used in different experiments (also for the RAMP controls). For example, 1+4 was shown to be best in Fig 1, but this is not tested in miniGs recruitment (Fig 2C) or P63 recruitment (Fig 4A) but it is shown for Nb35 recruitment (Fig 2B), fluorescent ligand binding (Fig 3), and IP1 accumulation (Fig 4C).

3. In the beta-arrestin knockout cells, where was MC4R localised? (And is there a potential role for beta-arrestin-independent endocytosis for constitutive MC4R recycling?) Similarly, can you speculate on whether MC4R is ever at the plasma membrane in the absence of MRAP2 (particularly given the localisation data in N7 cells)? This is worth discussing in the context of the influence of receptor location in the activation of signalling.

4. How does your homology model (and the structure of MARP1 with MC2R) compare with RAMP/GPCR interactions e.g. RAMP1 with CLR. Please add this to the discussion.

5. The data in Fig 3 suggests that MRAP2 may decrease the affinity of AF647-NDP-alpha-MSH for MC4R (i.e. tighter binding). If you have saturating binding data for AF647-NDP-alpha-MSH it would be useful to include this to quantify the change. Alternatively please cite previous publications where this has been investigated.

6. EC50 and EMax values from concentration-response curves:

- Please use pEC50 (and not EC50) values in the graphs in panels 1B, 2D, 4B, 4D, and for statistical analysis (as EC50 values are not normally distributed)
- Please check the EC50 calculations in Fig 8B, particularly for the H158R+mock condition – there seems to be a clear shift in the pEC50 in the grouped data shown in panel 8A, but this is not reflected in the EC50s displayed in 8B.
- EMax values for control conditions all sit at 100% (e.g. 6D) please confirm how the EMax values were obtained – if they were determined based on non-linear regression (e.g. in GraphPad prism) on each individual experiment, there should be more variation even if the data are normalised to the maximal response.

7. Please discuss the relative impact of MC4Rchim and MC4R-H158R on molecular brightness compared to previous publications from the team (Reininghaus et al 2022). In addition, please add some discussion about how the changes in MC4Rchim and MC4R-H158R could mechanistically impact receptor dimerization. In the cryo-EM structures of MC4R bound to Gs (Hadar et al Science 2021, Heyder et al Cel Res 2021), His158 seems to be directed inwards and closer to the G protein, rather than directed outwards to the membrane. It would be useful to indicate the position of H158 on the homology model in Fig 9.

8. In general, more detail is required in the methods regarding how data were quantified. For example, how was the Nb35 recruitment in Fig 2 quantified? Which laser lines were used for image acquisition? What wavelengths were detected? Were data analysed on a single cell basis (i.e. what defines a biological replicate)? Does the number of endosomes per cell (5D) refer to all endosomes or MC4R positive endosomes? How was molecular brightness calculated using ImageJ? How were data from the Epac FRET biosensor analysed & grouped (i.e. what defines a biological replicate)? (Also these experiments are not always analysed the same way – compared Fig 1A to Supp Fig 1C). How were ROIs for molecular brightness experiments selected (was this blinded)?

9. Results pg 7, line 172 "MRAP2 significantly increased the kinetics of nanobody recruitment" – please quantify these kinetics and show statistical analysis of the data.

10. Fig 1C and 5D – what is the difference in technical vs biological replicates? Why show the technical replicates here? Please only do statistical analysis on the biological replicates.

Minor points

- Please use IUPHAR nomenclature for the MC4R (the "4" should be subscript)
- Results pg 16 line 379 "G protein signaling cannot be solely due to the beta-arrestin-promoted endocytosis since a similar MRAP2-promoted leftward shift in Gs activation was observed in beta-arrestin1 and beta-arrestin2 double knock-out cells than in SL cells" – please rephrase as this relative shift in the knockout cells is smaller. While the EC50 in the presence of MRAP2 is unchanged, the EC50 in the absence of MRAP2 is different between the beta-arrestin KO and rescue cells.
- Fig 2D – change y scale to include all data points (missing value from control)
- What is the difference between Fig 2 & Supp Fig 3 (apart from magnitude of effect)?
- Fig 3B shows "n=1 independent transfection, 121 cells". While we acknowledge the high degree of variation across a single transfection, it would be useful to see how reproducible this is across independent experiments.
- Results pg 10 line 233 "No response was observed after alpha-MSH challenge in the absence of SNAP-MC4R expression, showing that the response is indeed MC4R-dependent (Supplementary Figure 5D)" – Supp Fig 5 is images of

receptor internalisation not p63RhoGEF activity

7. Results pg 10 line 248 "The observation that, in contrast to the p63RhoGEF BRET assay, no increase in the maximal response was observed most likely reflects a saturation of the IP1 response." Please rephrase to make clear whether the saturation is at the level of receptor stimulation or at the level of detection. The assay (IP-One Gq kit, CisBio) uses a standard curve to determine the amount of IP1 produced, however, the data are reported as HTRF Ratio (%). It is important to confirm that the IP1 produced by this experiment fits within the dynamic range (standard curve) of the assay. It is possible that the lack of effect of MRAP2 on the Emax could be due to saturation of the antibody detection method.

8. Supplementary Figure 5 legend: "In the right panel, cells expressing the receptor are further preincubated with 100 nM AgRP to suppress constitutive internalization." Please fix to reflect current layout of the panels.

9. Please check the brightness/contrast display of the images showing Rab5-CFP in Fig 5 – in most of these images the Rab5 signal appears cytosolic, as well as punctate.

10. Please state how you calculated the number of MC4R per um² (Supp Fig 9) based on the molecular brightness measurements. Does this difference with MRAP2 co-transfection account for the proposed shift from a dimer/oligomer to monomer receptor population? Do you have enough points to split the 1:1 vs 1:4 MRAP2 transfection conditions?

11. Methods, "Seeding and transfection" – please check that you clearly list which experiments used JetPrime transfection, and which used Lipofectamine 3000, and include PEI (used for the "BRET based assays") in this section.

Reviewer #3

(Remarks to the Author)

Reviewer #4

(Remarks to the Author)

In this study, the authors reported in a heterologous expression system that MRAP2 favors Gs- and Gq-dependent signaling of MC4R while inhibiting beta-arrestin-dependent signaling and internalization of MC4R. The authors also provided mechanistic insight into how MRAP2 interacts with MC4R to enhance the signaling. While the data are new and relevant to in vivo function of MRAP2, this study depends heavily on overexpression systems, and it is not clear whether these findings may apply in vivo where expression levels of receptors and signaling molecules should be much lower. Other specific points are listed below.

1. Please re-arrange supplementary figures in the order they appear in the text. It would be difficult for readers to go back and forth to navigate the supplementary figures as they read the manuscript. For example, in page 10 lines 235 and 246, supplementary figure 5D appears earlier than supplementary figure 2.

2. The main text contains too many details of experimental techniques, which makes it difficult to follow the manuscript. It would be better to include only essential information here and move details to methods.

3. It is currently not clear why the authors presented Figures 3B and 3C separately to explain a topic. This may be familiar to those in this field, but not to a general readership. Please resolve this issue.

4. It may be helpful to include paragraphs explaining in vivo relevance of the findings in discussion.

Version 2:

Reviewer comments:

Reviewer #1

(Remarks to the Author)

I am happy with the changes the authors have made to the manuscript. They have addressed most of my concerns.

Reviewer #2

(Remarks to the Author)

The author's have thoroughly responded to all our comments, and have included a large amount of new data and analyses which has further strengthened their manuscript.

Reviewer #3

(Remarks to the Author)

Reviewer #4

(Remarks to the Author)

In the revised manuscript, the authors have successfully addressed all of my previous concerns. One remaining issue is whether it is appropriate to mention that a recent study (Wyatt et al, 2025) was published while this manuscript was under revision (page 31, lines 723-724). Simply mentioning "a recent study" should be sufficient.

We are grateful to the reviewers for their insightful feedback. Our point-by-point rebuttal is found below:

Reviewer #1 (Remarks to the Author):

1. I would like to see the impact of the MRAP2 interaction MC4 receptor agonist binding. Are the affinities changed for the agonists if the MRAP2 is present? I think this is critical for the story here as it will impact downstream signalling.

We thank the reviewer for this important comment. To address it we have conducted two sets of additional experiments. **Results were amended in the revised Figure 3.**

First, we performed radioligand binding saturation experiments on membrane preparations from HEK293SL cells expressing MC4R alone or in the presence of MRAP2 or RAMP3 (as a negative control). The results are displayed in the image below, which has been added to the revised manuscript (**Figure 3C and D**). Co-expression of MRAP2 with MC4R (1+1) causes an increase in receptor level (B_{max}), albeit no sizeable change in affinity (pK_D).

Reviewers Figure 1 (Revised Figure 3C-D): Affinity and maximal binding parameters extracted from saturation radioligand binding experiments using [¹²⁵I]-NDP- α MSH on membrane preparations of HEK293-SL cells transfected with MC4R and empty vector (mock), MRAP2 or RAMP3 at 1+1 ratio. The right panel shows the affinity (as pK_D). The right panel show B_{max} normalized as a function of the value obtained for cells expressing only MC4R (mock). Data are displayed as mean \pm SEM of three independent experiments. Statistical analysis was performed using ordinary one-way ANOVA with Dunnett's multiple comparisons post-hoc test (* = p < 0.05). The data show that co-expression of MRAP2 promotes an increase in total MC4R level with no apparent change in affinity.

Both radioligand and single cell confocal experiments concur that MRAP2 expression correlates with increased receptor expression on the membranes. Indeed, we do see a positive correlation between the intensity counts from the SNAP-surface549-labelled SNAP-MC4R and from the EYFP-MRAP2 (normalized) in the confocal micrographs we took of HEK293 cells transfected with both constructs (now **Supplementary Figure 9A**). This observation is also in accordance with molecular brightness data showing that co-expression of MRAP2 at a 1+4 ratio increases MC4R density (**Supplementary Figure 9B**).

Reviewers Figure 2 (Supplementary figure 9): MRAP2 co-expression correlates with increase in cell-surface MC4R. (A) Labelling of SNAP-MC4R using a membrane impermeable dye (SNAP-surface549) allows correlating MC4R membrane abundance to overall expression of EYFP-MRAP2 (normalized), based on intensity counts from confocal micrographs of HEK293 cells co-transfected with SNAP-MC4R and EYFP-MRAP2 at a 1+4 ratio (in presence of 0,5 nM AF647-NDP- α -MSH). **(B)** MC4R density in the absence or in presence of MRAP2 1+4 co-expression.

Finally, we performed expanded single cell experiments using confocal microscopy to monitor the relative abundance of bound fluorescent ligand (AF647-labelled NDP- α -MSH), SNAP-MC4R and EYFP-MRAP2 upon application of two ligand concentrations, 500 pM (same as presented in the first version of the submitted manuscript) and now also 1 μ M (closely saturating concentration). The revised data, now included in the **Figure 3B**, show that MRAP2 increases the amount of ligand bound per receptor at a ligand concentration closer to the K_D (as calculated by radioligand binding).

Reviewers Figure 3 (Revised Figure 3A-B): Binding of NDP- α -MSH is increased HEK293 expressing MC4R and increasing amounts of MRAP2. (A) Confocal maximum intensity projections of representative HEK293 cells transfected with either SNAP-MC4R + mock or SNAP-MC4R + MRAP2 (1+4), seeded on the same coverslip and then labelled using SNAP-surface549. The cells were then incubated for 5 minutes with either 500 pM or 1 μ M Alexa647-NDP- α -MSH and imaged. Contrast is set to the same values in each channel. Maximum intensity projections were constructed in FIJI before running Cellpose-based cell segmentation to extract individual cells intensity values. Scale bar is 10 μ m. **(B)** Chart of the normalised NDP- α -MSH ligand signal (to SNAP-MC4R receptor expression) per cell for 500 pM ligand (red circles, n=5 independent transfections, 226 cells) and 1 μ M ligand (n=2 independent transfections, 45 cells). Shading indicates MRAP2 positive cells, as determined by visual inspection of confocal micrographs.

The fact that when labelling cells with 0.5 nM Alexa647-NDP- α -MSH in absence of MRAP2, very little signal from the fluorescent ligand is observed, as opposed to cells expressing MRAP2 (as detected by EYFP signal) (**Reviewers Figure 3A,B**) suggests an effect of MRAP2 on ligand binding affinity in intact cells. This effect is saturable, since when labelling the cells with 1 μ M Alexa647-NDP- α -MSH, also MRAP2 negative cells display substantial binding.

Overall, the increase in receptor level seen with both methods (radioligand binding on membranes and microscopy on intact cells) would be compatible with the increased signalling potency and or efficacy for all pathways observed in the presence of MRAP2. This effect would be further strengthened by the increase in binding affinity of NDP- α -MSH detected in intact cells.

To accommodate the reviewers request, the text in the results section **MRAP2 expression favours Gs recruitment to the MC4R and enhances α -MSH binding at the receptor** has been changed as follows:

*These results are mirrored when measuring ligand binding. Using a single cell-based fluorescence assay in conjunction with a fluorescently tagged NDP- α -MSH, we could observe that intact cells transfected with MRAP2 (1+4) bind significantly more ligand than the mock transfected counterparts (**Figure 3A**) and that the number of NDP- α -MSH molecules bound, normalized by the number of receptors expressed, increases as a function of the expression level of MRAP2 in transiently transfected cells (**Figure 3B**) at concentrations of fluorescently tagged NDP- α -MSH of 0.5 nM. However, at saturating concentrations of the ligand (1000 nM), binding appears independent of MRAP2 expression level. When conducting radioligand binding on membranes from HEK293 cells using [125 I]-NDP- α MSH, we observed an MRAP2-independent affinity **Figure 3C**. On the other hand, Bmax values were increased in membranes from cells transfected with MRAP2 (1+1), suggesting an increase of the number of receptors present at the plasma membrane (**Figure 3D**).*

2. While there has been effort to use to ratios of MRAP2 to MC4 receptor expression, I would like to see a situation assay confirming that the amounts used are appropriate. This is critical since how can the investigators be confident that ALL cells express both proteins.

The reviewer raises an important point, and we are aware that cells might express at variable rates or not-at-all each of the plasmids transfected. In the confocal microscopy experiments in Figure 3 in the original submission, we showed that transfected cells express a variable amount of both MC4R and MRAP2. In those experiments, cells were transfected with the receptors, but only half were transfected with MRAP2 before being mixed. The ligand/receptor ratio is then plotted against MRAP2 expression, as monitored by EYFP fluorescence, so that the level of MRAP2 expression for each cell is known by definition. This setting allowed us to observe a variability in EYFP-MRAP2 expression and its positive correlation with MC4R cell-surface expression, as now displayed in **Supplementary Figure 9A**.

In relation to the cell-based assays shown (e.g. in **Figure 1** of the manuscript), plate reader readouts are averaged over tens of thousands of cells, and we expect that most cells co-transfected with MRAP2 and MC4R express both proteins as we do see an impact of the co-expression of MRAP2 on the function of MC4R. Furthermore, co-expression was confirmed for some assays by direct inspection, since SNAP-MC4R could be labelled with a membrane impermeable SNAP dye and MRAP2 expression could be inferred from the presence of mRuby2 fluorescence, as both proteins were subcloned within the same pViro2 dual expression vector, which allows their concomitant and constitutive expression from two different composite

promoters. Microscopy allows direct and simultaneous detection of MC4R (SNAP dye) and MRAP2 (mRuby reporter), showing that both are present, and at which levels.

To address these changes, the following text has been introduced in the results section ***MRAP2 expression alters the MC4R oligomeric equilibrium towards a monomeric state***

Furthermore, MRAP2 appears to increase the average number of cell surface expressed MC4Rs in a concentration dependent way (Supplementary Figure 9A), leading to a significantly increased receptor expression levels in cells co-expressing MRAP2 (1+4) (Supplementary Figure 9B), in line with the observations reported in Figure 3C and Figure 5.

3. To this end, have the investigators considered the use of some form of molecular glue to ensure that the MRAP2 and MC4R are "forced" together. While this might impact signalling it will give confidence in the interaction.

Our investigation originates from the observation that in our experiments MRAP2 co-expression does affect MC4R function, so we have confidence in the interaction. This is true both in averaged experiments (such as those shown in **Figures 1, 4, 5E, 6 and 8** in the manuscript, as well as in assays that yielded single-cell results (**Figures 2, 3, 5, 7** in the manuscript). While forcing the interaction may shed some light on effects such a sequestered pool of MRAP2s, we are afraid that forcing the interaction may also lead to unphysiological outcomes, which would not further our understanding of the MRAP2-MC4R interaction.

However, in recognition of the point raised by the reviewer, we have conducted a set of direct molecular interaction assays, to inspect directly the interaction between the two proteins. The assays were performed using a variation of the fluorescence cross-correlation spectroscopy approach, a technique that has been used for decades to monitor molecular interactions, including in the GPCR field, by means of visualizing co-diffusion through a confocal excitation volume (Teichmann et al. 2012), and reviewed in (Scarselli et al. 2016).

In this implementation we generated spatial-temporal diffusion maps, comparing the MC4R and MRAP2 pair to a negative control (β 1-AR and MRAP2, not known to interact), and a positive control, the CXCR4 dimer (Isbilir et al. 2020), (Ward et al. 2021). GPCRs are labeled with a cell membrane impermeable SNAP dye, whereas MRAP2 is labeled with EYFP. The results clearly prove direct molecular interaction between MRAP2 and MC4R, as a cross-correlation function with a sizable amplitude is observed (**Supplementary Figure 10D**), albeit to a lesser degree of the positive control. Diffusion rates of the complexes, of the order of 0.3-0.4 $\mu\text{m}^2/\text{s}$ are in line with what is widely reported for GPCRs.

These data are presented as a new supplementary figure 10 and are mentioned in result section ***MRAP2 expression alters the MC4R oligomeric equilibrium towards a monomeric state***. and the text in this section has been amended as follows:

*Based on these results, we validated the direct interaction between MC4R and MRAP2 using a related fluorescence fluctuation spectroscopy method to Molecular Brightness, namely Fluorescence Cross-Correlation Spectroscopy (FCCS) ((Teichmann et al. 2012), and reviewed in (Scarselli et al. 2016)). FCCS data (**Supplementary Figure 10**) shows unambiguously that SNAP-*

MC4R co-diffuses with EYFP-MRAP2, in a fashion comparable to the constitutively dimeric CXCR4 (Isbilir et al. 2020). On the contrary, a negative control monitoring EYFP-MRAP2 co-transfected with the SNAP- β 1-AR, did not display any co-diffusion.

Reviewers Figure 4 (Revised Supplementary Figure 10): Molecular interaction between MC4R and MRAP2 probed using Spatial-temporal Fluorescence Cross-Corelation Spectroscopy (ccSTICS). (A) Average ccSTICS function of the interaction between SNAP-MC4R, labeled with Snap-surface Alexa647, and EYFP-MRAP2, displaying the presence of co-diffusion (n=3 independent transfections, 15 cells). **(B)** Average ccSTICS function of the interaction between SNAP- β 1-AR, labeled with Snap-surface Alexa647, and EYFP-MRAP2, displaying the absence of any co-diffusion. (n=2 independent transfections, 13 cells) **(C)** Average ccSTICS function of the interaction between SNAP-CXCR4 protomers, stoichiometrically labeled with Snap-surface Alexa549 and Alexa647, displaying the presence of co-diffusion consistent with the dimeric nature of this receptor. (n=2 independent transfections, 19 cells). **(D)** Resulting cross-correlation functions, illustrating comparable diffusion times for the CXCR4 homodimer ($D=0.34 \mu\text{m}^2/\text{s}$) and the MC4R-MRAP2 heterodimer ($0.43 \mu\text{m}^2/\text{s}$).

4. RAMP proteins have been shown to impact signalling (e.g. CLR, GCGR, AMYR) through the C-terminal tails. Did the investigators consider remove the MRAP2 tail to see if effects are removed.

While we recognise that extensive mutagenesis of MRAP2, including deletions of specific parts, could provide further insight into the molecular interaction with MC4R, it is important to note that unfortunately none of the previously postulated models of MRAP2 (e.g. in our study) nor the determined structure of MRAP1 with MC2R (Luo et al. 2023), include the C-terminus. This makes predictions about the interaction between the (most likely disordered) C-terminus of MRAP2 and any GPCR rather speculative, thus, we did not specifically address this question.

It is worth mentioning that another research group has addressed the question of the interface of interaction between MC4R and MRAP2 by mutagenesis while our manuscript was under revision

(Wyatt et al. 2025). This study has indeed shown that some mutations in the C-terminus can alter MC4R internalization.

5. RAMP3 has a PDZ domain linked to trafficking. Does MRAP2 have the same effect?

This is an interesting issue. MRAP2 doesn't have a PDZ domain according to <http://modpepint.informatik.uni-freiburg.de/> and <https://pow.baderlab.org/>. A comment about this has now been included in the **Discussion** section of the manuscript.

This could be either be attributed to a decrease in endocytosis or to an increased in cell-surface targeting of MC4R from the secretory pathway, akin to what observed for MRAPs and the MC2R (Webb and Clark 2010). We shall note here that, with respect to trafficking, MRAP2 -as opposed to RAMP-3, is not predicted to display a PDZ domain(Bomberger et al. 2005).

6. Have the investigators considered the use of BRET to show MRAP2 and MC4R interactions - both on the cell surface and intercellular?

On MC4R-MC4R homo-oligomerisation:

We think the molecular brightness experiment (**Figure 7** in the manuscript) is more robust than the BRET assay to show homo-oligomeric state of the MC4R, since a change in BRET can also be underpinned by a reorganization of the homodimer rather than a monomerization. Moreover molecular brightness readouts (conducted in confocal mode) are (more) selective of the plasma membrane of the cell as opposed to luminescence readouts that lack any optical sectioning.

On MC4R-MRAP2 hetero-oligomerisation:

We conducted new nano luciferase complementation assays. The co-expression of MC4R-LgBit and MRAP2-Smbit resulted in a substantial level of luminescence with respect to the control with untagged MRAP2, supporting the interaction between MRAP2 and MC4R.

Reviewers Figure 5: Hetero-dimerization assessed by Nano luciferase protein complementation assay between MC4R and MRAP2

We also believe that the cross-correlation experiments conducted to answer Point 3 of the reviewer also address this question, and to this purpose we have generated a new **Supplementary Figure 10**. It should be noted that the cross-correlation spectroscopy

experiments were conducted in confocal mode at the plasma membrane validating that the heterodimer exist specifically at the cell surface.

7. I find the homology section speculative. I think this section would benefit from MD simulations which are very intensive. If the investigators which to keep this section in the manuscript, I think it would be better in the supporting information as without MD simulation I am not sure it is adding much to the study.

An extended MD simulation, as noted by the reviewer, is certainly attractive, but cannot be completed within the specified time frame and was also outwit the scope of this work. The reviewer also noted that the initial model is too speculative, and this is unfortunately also true for an extended MD simulation, as there is still no adequate experimental structure for MRAP2 in complex with MC4R and NDP-a-MSH. This is of course the core of the problem. An extended MD simulation is only useful based on more complete complex experimental determined models that also include the C-terminus of MRAP2, or if the study were performed solely to identify interaction points between the two proteins.

However, we believe that the homology model shown here, which is based on an existing MC2R-MRAP1 structure, helps to visualise and understand the described interplay between these two membrane-spanning proteins. The overall structure will certainly be similar, but e.g., specific interactions or the role of the C-terminus and the resulting implications are not covered by this and are thus not discussed in our manuscript. Several aspects of our model are of interest to answer potential questions regarding the heterodimer, such as a possible direct influence on structural aspects of MC4R activation in the transmembrane region or extracellularly.

However, if the reviewers believe this is more appropriate, we are open to moving parts related to modelling and certain derived information to the Supplementary Material section.

Reviewers #2 and 3:

1. We think it is important to confirm that the amount of MC4R protein in the transfected cells is consistent across the different transfection conditions (i.e. does switching from co-transfection with empty vector, to 1+1 MRAP2 or 1+4 MRAP2 impact the amount of MC4R produced?). A change in the amount of MC4R expressed in the cells could also impact EC50 and/or EMax values.

We thank the reviewer for this important comment. By combining three assays (confocal imaging, molecular brightness and radioligand binding), we observed that the level of cell surface MC4R expression is increased upon co-expression of MRAP. **We kindly refer you to the answer of Point 1 of Reviewer 1.**

Also, as the reviewer points out, we are aware that increased MC4R expression would affect EC50 and Emax. We indeed observed during optimization stages a leftward shift of the EC50 and an increase in Emax as more MC4R is transfected into the cells in all assays.

Similarly, please confirm that transfecting increasing amounts of MRAP2 results in increased expression of MRAP2 protein (this is important to confirm the proposed saturable effect on EC50 suggested for panel 1B).

This effect of saturation of the effect of MRAP2 over-expressions on the EC50 was seen in all signaling assays conducted (**Figures 1B, 2D, 10**), although MRAP2 levels in these experiments were not directly monitored. In parallel experiments, we monitored the expression of GFP10-MRAP2 in cells also expressing a MC4R construct. As can be seen in the **Reviewers Figure 6** below, the expression of MRAP (monitored by the fluorescence signal) increased linearly with the amount of GFP10-MRAP2 transfected. The range of plasmid transfected in those parallel experiments was greater than in the signaling experiments used in the manuscript, suggesting that an increase in MRAP2 most likely occurred with increasing amount of plasmid transfected in these experiments. This also implies that the reason for the saturation rather comes from another limiting factor.

Reviewers Figure 6: Mean total fluorescence of increasing amounts of GFP10-MRAP2. GFP10: excitation 400nm / emission 515 (cutoff 515). Data are only one experiment, represented as mean \pm SD of technical triplicate.

2. Part of the rationalisation for the current study was to more precisely characterise the MRAP2/MC4R interaction i.e. Results pg 5, line 118 “... wide range of transfection ratios... highly variable outcomes are reported, without proper explanation for their selection”

Given this statement, it seems important to clearly rationalise why different transfection ratios are used in different experiments (also for the RAMP controls). For example, 1+4 was shown to be best in Fig 1, but this is not tested in miniGs recruitment (Fig 2C) or P63 recruitment (Fig 4A) but it is shown for Nb35 recruitment (Fig 2B), fluorescent ligand binding (Fig 3), and IP1 accumulation (Fig 4C).

We thank the reviewer for this comment. Our selection of the 1+1 and 1+4 ratios stems from the reported observation that mRNA levels in the paraventricular hypothalamic nucleus are of the order of 1+2. Expression ratios of 1+1 and 1+4 were therefore chosen as boundaries encompassing this reported ratio.

Protein	nTMP	MC4R:MRAP2
MC4R	3.6	1:3
MRAP2	11.5	

Supplementary Table 1: The normalized gene expression values (nTMP) representing the RNA expression of the MC4R and MRAP2 in the paraventricular nucleus of the hypothalamus (The Human Protein Atlas, 2023)(Uhlen et al. 2015). The values obtained give an expression ratio for MC4R to MRAP2.

The different ratios used was in some instances imposed by the specific assay conditions that limited the amount of DNA that could be transfected.

We will remove the statement (*In the literature, a wide range of transfection ratios of MRAPs:GPCRs (1:1 to 1:20) with highly variable outcomes are reported, without proper explanation for their selection*) on page 5, line 118 of the original submission, to avoid raising unnecessary critique of previous studies, replacing it with the statement (**RESULTS MRAP2 expression increases Gs-mediated cAMP accumulation in the basal and ligand-activated states of MC4R**):

Our selection of the 1+1 and 1+4 ratios stems from the reported observation that mRNA levels in the paraventricular hypothalamic nucleus are of the order of 1+2 (Uhlen et al. 2015). Expression ratios of 1+1 and 1+4 were therefore chosen as boundaries encompassing this reported ratio.

3. In the beta-arrestin knockout cells, where was MC4R localised? (And is there a potential role for beta-arrestin-independent endocytosis for constitutive MC4R recycling?) Similarly, can you speculate on whether MC4R is ever at the plasma membrane in the absence of MRAP2 (particularly given the localisation data in N7 cells)? This is worth discussing in the context of the influence of receptor location in the activation of signalling.

We thank the reviewers for this comment. In the β -arrestin KO cells that we used, we can confirm the presence of MC4R at the plasma membrane from separate BRET assays. As can be seen in the figure below, cell surface expression of MC4R (as detected by the BRET between MC4R-Rluc and rGFP-CAAX) was similar in cell KO for β -arrestin or in which β -arrestin was reintroduced. This is in contrast with the low level of surface receptors observed for the R165W mutant form of MC4R known to be retained inside the cells (Rene et al. 2021). This indicates that MC4R does not require β -arrestin to reach the cell surface and that reintroduction of β -arrestin in the KO background does not affect the expression.

Reviewers Figure 7: MC4R-RlucII localization to rGFP-CAAX in β -arrestin KO cells, in presence or absence of beta-arrestin1&2 reintroduction. Rescue was performed with 50 ng of each β -arrestin1 and 50 ng of β -arrestin2.

We did show previously that MC4R acute agonist-promoted internalization is β -arrestin dependent as short time endocytosis is completely blocked in β -arrestin 1 and 2 double knock-out cells (Brouwers et al. 2021). A statement concerning the β -arrestin dependency of the MC4R endocytosis has been added to the manuscript (results section **Overexpression of MRAP2 decreases β -arrestin2 recruitment to the receptor**):

*Despite the possible link between the increase in G protein signaling and the decrease β -arrestin2 recruitment and receptor endocytosis, we shall point out that the MRAP2-mediated increase in G protein signaling cannot be solely due to a reduction in β -arrestin-promoted endocytosis, since a leftward shift in Gs activation was also observed in β -arrestin1 and β -arrestin2 double knock-out cells when transfected with MRAP2 (**Supplementary Figure 7**).*

We respectfully suggest that investigating whether β -arrestin is involved in constitutive endocytosis falls outside of the scope of this paper, in particular since the potentiating effect of MRAP2 on MC4R Gs and Gq-mediated signaling was not abolished in β -arrestin KO cells.

4. How does your homology model (and the structure of MRAP1 with MC2R) compare with RAMP/GPCR interactions e.g. RAMP1 with CLR. Please add this to the discussion.

We thank the reviewer for this interesting question. We have already analysed known RAMP-GPCR complexes in detail (Nemec et al., *PNAS* 119 (32) e2122037119, <https://doi.org/10.1073/pnas.2122037119> (2022); Fig. 7a, SI_Fig. S12) and have recently made some basic internal comparisons between RAMPs and MRAPs: firstly, a sequence comparison between MRAPs and RAMP1 or RAMP2 quickly reveals that these proteins are fundamentally different, even if their generally assumed effects on GPCRs may ultimately show some overlap.

Reviewers Figure 7: Sequence comparison between MRAPs and RAMP1 or RAMP2.

Furthermore, the previously assumed interfaces of RAMP-GPCR and MRAP2-MC4R in the transmembrane region are completely different based on the fact that MRAP1 and MC2R interact differently than RAMPs and CRLR (see **Reviewers Figure 8**). In conclusion, it can be clearly stated that the mechanisms of action of MRAP and RAMP at the extracellular vestibule are fundamentally different with regard to ligand binding modification.

In view of this and the fact that RAMPs to our knowledge have so-far been shown to only interact with class B1 receptors, and not with class A GPCRs, our study focuses on the description of MRAP-GPCR interactions. While we agree that a comparison between activity-modulating protein classes, MRAPs and RAMPs would be interesting, we believe it falls outside the scope of the current work.

Reviewers Figure 8: MC2R (cyan) and MRAP1 (blue) (PDB ID 8gy7) have an interaction interface that is completely different than that of RAMP1(red) and CLR (beige) (PDB ID 7knu).

5. The data in Fig 3 suggests that MRAP2 may decrease-increase the affinity of AF647-NDP-alpha-MSH for MC4R (i.e. tighter binding). If you have saturating binding data for AF647-NDP-alpha-MSH it would be useful to include this to quantify the change. Alternatively please cite previous publications where this has been investigated.

We thank the reviewer for this comment. We kindly refer to the answer to **Point 1 of Reviewer 1** in which we present the new experiments done to assess the effect of MRAP2 on the ligand affinity and cell surface abundance of MC4R.

As far as prior literature is concerned, we would like to report a previous work where competition binding assays were conducted against the zebrafish zMC4R, expressed recombinantly in intact HEK293 cells(Sebag et al. 2013). These data (**Reviewers Figure 9**) show, in agreement with our radioligand binding measurements with human MC4R and MRAP2, changes in Bmax.

Reviewers Figure 9: Displacement of radioligand labeled NDP-α-MSH of zMC4R in HEK2993 cells (adapted from (Sebag et al. 2013)).

To account for these aspects, we have introduced in the **Discussion** section the following paragraph:

We have indeed observed an increase in MC4R cell-surface expression upon co-expression of MRAP2 in HEK2993, which positively correlates with the amount of MRAP2 detected in the cells by both microscopy and radioligand binding (Figure 3C, Supplementary Figure 9). These results are in line with previous observations from radioligand studies on the zebrafish MC4R(Sebag et al. 2013).

6. EC50 and EMax values from concentration-response curves:

a. Please use pEC50 (and not EC50) values in the graphs in panels 1B, 2D, 4B, 4D, and for statistical analysis (as EC50 values are not normally distributed)

We thank the reviewer for this comment, we have implemented their recommended correction in the revised manuscript.

b. Please check the EC50 calculations in Fig 8B, particularly for the H158R+mock condition – there seems to be a clear shift in the pEC50 in the grouped data shown in panel 8A, but this is not reflected in the EC50s displayed in 8B.

To doublecheck, the reviewer is right. This was due to erroneously copying an older version of the figure panel, this has now been corrected.

c. EMax values for control conditions all sit at 100% (e.g. 6D) please confirm how the EMax values were obtained – if they were determined based on non-linear regression (e.g. in GraphPad prism) on each individual experiment, there should be more variation even if the data are normalised to the maximal response.

Emax was normalized to the top asymptote of the mock condition in each experiment. Thus, the average of the maximal response (Emax) will be 100% for the mock condition.

7. Please discuss the relative impact of MC4Rchim and MC4R-H158R on molecular brightness compared to previous publications from the team (Reininghaus et al 2022). In addition, please add some discussion about how the changes in MC4Rchim and MC4R-H158R could mechanistically impact receptor dimerization. In the cryo-EM structures of MC4R bound to Gs (Hadar et al Science 2021, Heyder et al Cel Res 2021), His158 seems to be directed inwards and closer to the G protein, rather than directed outwards to the membrane. It would be useful to indicate the position of H158 on the homology model in Fig 9.

Reviewers Figure 10: BRET experiments to study the oligomerisation of various MC4R mutants, from Reininghaus et al 2022.

We thank the reviewer for their comment. Our past data that assessed dimerisation using BRET on MC4R mutants, indicated a clear break of the dimeric fingerprint for the MC4Rchim (indicated as Chim7) and for the H158R mutant (**Reviewers Figure 10**). These results are fully in agreement with the brightness experiments reported in the present manuscript (**Figure 7**). Overall, the combined data support that MC4R-chim and H158R have a very similar fingerprint, though MC4R-H158R has a bit more variability.

We have amended the **Discussion** section of the manuscript to highlight that H158R is in ICL2, which lies in one of the proposed homodimerization interface, i.e. the one between ICL2 and TM4 (Kleinau et al. 2020) as follows:

Towards this purpose we tested the naturally occurring homo-dimerization-deficient MC4R-H158H mutation. This is located in the putative MC4R interaction interphase between ICL2 and TM4. We further tested a homo-dimerization-deficient chimeric variant (MC4Rchim) in which parts of MC4R between ICL2 and TM4 were exchanged with CB1R. We observed that that both MC4R-H158R and MC4Rchim display Gs signaling comparable to wild type MC4R in absence of MRAP2 and enhanced Gs signaling in the presence of MRAP2 (Figure 8).

8. In general, more detail is required in the methods regarding how data were quantified. For example, how was the Nb35 recruitment in Fig 2 quantified? Which laser lines were used for image acquisition? What wavelengths were detected? Were data analysed on a single cell basis (i.e. what defines a biological replicate)? Does the number of endosomes per cell (5D) refer to all endosomes or MC4R positive endosomes? How was molecular brightness calculated using ImageJ? How were data from the Epac FRET biosensor analysed & grouped

(i.e. what defines a biological replicate)? (Also these experiments are not always analysed the same way – compared Fig 1A to Supp Fig 1C). How were ROIs for molecular brightness experiments selected (was this blinded)?

We have amended the methods section to include this information. Specifically, to the Nb35 recruitment experiments, we used an excitation wavelength of 514 nm to quantify Nb35 recruitment and an excitation of 633 nm to quantify receptor on the membrane. Data were measured across single cells (technical replicates, generated across multiple transfections). We note that this information is reported in **Figure 2** caption. We have specified in the **Figure 5D** caption that we refer to MC4R positive endosomes.

We have added an updated methods section, expanding the information about Brightness analysis that was originally provided as follows:

Polygonal region of interest (ROI) selection was implemented and areas with inhomogeneous fluorescence distribution were avoided.

ROIs for molecular brightness were selected based on homogeneous fluorescence intensity, from all cells selected for analysis. We followed the strategy presented in recent publications from our Team ((Annibale and Lohse 2020), (Isbilir et al. 2021)) which explain in more in detail the protocol we followed.

Concerning EPAC FRET biosensor data, we use two different FRET EPAC biosensors: the EPAC-S^{H158} (used in **Figure 1** and **Figure 8A**) and the sCFP3A-EPAC-venus (**Supplementary Figure 1C**). For the EPAC-S^{H158} biosensor, experiments were conducted in a plate reader, and biological replicates were independently transfected plates. This information is contained in the caption of **Figure 1** and subsequent Figures that report on FRET biosensing. For the sCFP3A-EPAC-venus FRET biosensor, experiments were also conducted in 96-well plates, and biological replicates represent independent transfections (different days of transfection) where each condition has only one well (no technical replicate).

Regarding differences in normalisation and analysis, as the results come from four different laboratories, sometimes there are minor differences in the way curves are presented or normalised, albeit we always tried to explain how this is done in each experiment in the Figures captions. We believe that the overall agreement between experiments replicated across different labs is an important strength of our manuscript. Specifically, **Figures 1 and 8A** are done with the EPAC-SH158 FRET biosensor and data were normalized to the system maximum obtained in presence of IBMX and forskolin, whereas **Supplementary Figure 1C** was done with the sCFP3A-EPAC-venus biosensor, and data were not normalized to an external system maximum.

9. Results pg 7, line 172 “MRAP2 significantly increased the kinetics of nanobody recruitment” – please quantify these kinetics and show statistical analysis of the data.

Data were not fit to a specific kinetic model. Qualitatively, and quantitatively, it is clear that, since traces are aligned to the time point of addition of the stimulus, the MRAP2 traces reach saturation when the MC4R only traces are at approximately 40% of maximum. The data shown is the average over several biological replicates.

However, for the benefit of the reviewer, the pool of non-normalized traces is shown below, where each line represents the fluorescence intensity collected in TIRF at the membrane from the Nb35-EGFP channel, after baseline subtraction. The data in **Figure 2** were further normalised, cell by cell, and averaged. **Reviewers Figure 11** below illustrates clearly that there is an enhanced recruitment (increased saturation value) of Nb35-EYFP to the membrane. We have further calculated the k_{on} from the data in **Figure 2B** of the manuscript, by fitting a single exponential recovery, obtaining the following values.

Reviewers Figure 11: (A) individual Nb37 membrane recruitment traces after NDP- α MSH challenge at time $t=100$ s for HEK293 cells expressing MC4R co-expressed with MRAP2 (green traces, $n=42$) or a mock plasmid (gray traces, $n=27$). Overlaid in bold mean and sem. (B) saturation values of the fluorescence for the two conditions. (C) kinetic time constants (in s) obtained from double exponential fitting of the traces shown in A.

It is evident that the amplitude of recruitment of Nb35 to the membrane for the cells co-expressing (1+4) MRAP2 (green in the chart) is significantly enhanced.

We have inserted in the results section **MRAP2 expression favours Gs recruitment to the MC4R and enhances α -MSH binding at the receptor** the following statement:

Notably, MRAP2 significantly increased the kinetics of the nanobody recruitment, effectively doubling the kinetic on-rate, suggesting that Gs recruitment to the receptor and its activation are indeed favored by the action of MRAP2.

10. Fig 1C and 5D – what is the difference in technical vs biological replicates? Why show the technical replicates here? Please only do statistical analysis on the biological replicates.

In Figure 1C technical replicates are different plates coming from the same transfection. As indicated in the caption of **Figure 1A**, each biological replicate has three technical replicates. We note that, when conducting analysis on technical replicates, as in **Figure 1C**, the standard deviation (as opposed to SEM) is shown.

Minor points

1. Please use IUPHAR nomenclature for the MC4R (the “4” should be subscript)

We thank the reviewer(s) for pointing this out. We will definitely address this matter if the manuscript moves to production.

2. Results pg 16 line 379 “G protein signaling cannot be solely due to the beta-arrestin-promoted endocytosis since a similar MRAP2-promoted leftward shift in Gs activation was observed in beta-arrestin1 and beta-arrestin2 double knock-out cells than in SL cells” – please rephrase as this relative shift in the knockout cells is smaller. While the EC50 in the presence of MRAP2 is unchanged, the EC50 in the absence of MRAP2 is different between the beta-arrestin KO and rescue cells.

Since the reviewer correctly pointed out that we should present the data as pEC50 and non EC50, the figures and the statistical analysis were amended. A statistically significant MRAP2-promoted decrease in pEC50 was observed in all cell background (HEK293-SL, HEK293-SL with β -arrestin KO, and HEK293-SL with β -arrestin KO and β -arrestin rescue) indicating that this effect on the MC4R signaling potency cannot be due to loss of β -arrestin-promoted endocytosis. As can be seen in the new **Supplementary Figure 7** (also pasted below), not only was the decrease in potency was observed in all cases, but the difference in pEC50 was also not statistically different between the different cell backgrounds.

To avoid confusion, the sentence referred-to by the reviewer was reformulated as follows:

we shall point out that the MRAP2-mediated increase in G protein signaling cannot be solely due to a reduction in β -arrestin-promoted endocytosis, since a leftward shift in Gs activation was also observed in β -arrestin1 and β -arrestin2 double knock-out cells when transfected with MRAP2 (Supplementary Figure 7).

Reviewers figure 12 (revised Supplementary Figure 7): MiniGs recruitment in β -arrestin1 and β -arrestin2 double knock-out cell. (A) Average concentration-response curves of Rluc8-miniGs recruitment to the plasma membrane localization sensor rGFP-CAAX upon MC4R stimulation with α -MSH for 45 minutes, in the presence or absence of MRAP2 or RAMP3, in transiently transfected β -arrestin1 and β -arrestin2 double knock-out cells. **(B)** β -arrestin1 and β -arrestin2 expression rescued the phenotype observed in related SL cells (not directly parental cells, as KO cells were maintained in FBS medium, while SL were maintained in NCS medium). **(C)** Data in HEK293SL from Figure 2C are shown again for easier comparison. Normalized data are expressed as mean \pm SEM of three independent experiments. **(D)** pEC₅₀ values from the miniGs recruitment BRET experiments in double KO cells, in absence (KO) or presence of the reintroduction of β -arrestins (rescue), expressed as mean \pm SEM of three independent experiments. pEC₅₀ in HEK293-SL cells from Figure 2D are reported for comparison (SL). **(E)** Difference in pEC₅₀ as compared to mock condition are displayed. Statistical analysis was performed using two-way ANOVA with Turkey's multiple comparisons post-hoc test (** = $p < 0.01$; **** = $p < 0.0001$). Not all statistical results are shown for better clarity.

3. Fig 2D – change y scale to include all data points (missing value from control)

Since all the EC₅₀ data were corrected to pEC₅₀, as pointed out in the response to the previous reviewer, this issue has been resolved.

4. What is the difference between Fig 2 & Supp Fig 3 (apart from magnitude of effect)?

In **Supplementary Figure 3** Gs is labeled with the fluorescent protein cerulean. In **Figure 2** it is unlabeled Gs. We first optimised our assay using cerulean tagged Gs. Once we determined our ability to see an effect by optimised transfection conditions, an untagged construct was used in order to minimise any disturbing effect due to the cerulean fusion, which may reduce, as correctly pointed out by the reviewer, the magnitude of the effect.

5. Fig 3B shows “n=1 independent transfection, 121 cells”. While we acknowledge the high degree of variation across a single transfection, it would be useful to see how reproducible this is across independent experiments.

This has now been reproduced across three independent experiments, and the caption amended accordingly.

6. Results pg 10 line 233 “No response was observed after alpha-MSH challenge in the absence of SNAP-MC4R expression, showing that the response is indeed MC4R-dependent (Supplementary Figure 5D)” – Supp Fig 5 is images of receptor internalisation not p63RhoGEF activity.

We thank the reviewer(s) for pointing this out. This has been corrected by citing the right figure.

7. Results pg 10 line 248 “The observation that, in contrast to the p63RhoGEF BRET assay, no increase in the maximal response was observed most likely reflects a saturation of the IP1 response.” Please rephrase to make clear whether the saturation is at the level of receptor stimulation or at the level of detection. The assay (IP-One Gq kit, CisBio) uses a standard curve to determine the amount of IP1 produced, however, the data are reported as HTRF Ratio (%). It is important to confirm that the IP1 produced by this experiment fits within the dynamic range (standard curve) of the assay. It is possible that the lack of effect of MRAP2 on the Emax could be due to saturation of the antibody detection method.

A Standard curve was run and analysed for optimization of this assay and the response fits within the dynamic range. The readout of raw data for this is the HTRF ratio which is inversely proportion to the IP1 response. The data reported in the graph has been presented as normalized response to maintain homogeneity with other figures that is why shown as % response. It was incorrectly labelled as HTRF Ratio (%) which has now been amended. It is possible that the difference observed between the p63RhoGEF BRET assay and the IP-One Gq readout, which operates downstream, is the fact that we are operating in a spare receptor mode, achieving saturation in IP1 response without activating all the receptors/G proteins.

Reviewer figure 13: Standard curve for IP-1 assay.

8. Supplementary Figure 5 legend: “In the right panel, cells expressing the receptor are further preincubated with 100 nM AgRP to suppress constitutive internalization.” Please fix to reflect current layout of the panels.

We thank the reviewer(s) for pointing this out. This has been corrected.

9. Please check the brightness/contrast display of the images showing Rab5-CFP in Fig 5 – in most of these images the Rab5 signal appears cytosolic, as well as punctate.

Done

10. Please state how you calculated the number of MC4R per μm^2 (Supp Fig 9) based on the molecular brightness measurements. Does this difference with MRAP2 co-transfection account for the proposed shift from a dimer/oligomer to monomer receptor population? Do you have enough points to split the 1:1 vs 1:4 MRAP2 transfection conditions?

We thank the reviewer for this observation. In molecular brightness experiment, the output is the brightness (photons/exposure/protomer) of the molecular species under investigation. If we divide the total intensity collected (total photons/exposure) by the brightness, we obtain the total number of emitters. In the case of a dimeric population (MC4R+mock), we multiplied this by 2 in order to obtain the number of protomers (MC4R/ μm^2). We conducted most of our molecular brightness experiments at a 1:4 ratio, so we would not have 1:1 data to include here with sufficient statistical robustness.

11. Methods, “Seeding and transfection” – please check that you clearly list which experiments used JetPrime transfection, and which used Lipofectamine 3000, and include PEI (used for the “BRET based assays”) in this section.

We thank the reviewer(s) for pointing this out. This has been corrected.

Reviewer #4 (Remarks to the Author):

In this study, the authors reported in a heterologous expression system that MRAP2 favors Gs- and Gq-dependent signaling of MC4R while inhibiting beta-arrestin-dependent signaling and internalization of MC4R. The authors also provided mechanistic insight into how MRAP2 interacts with MC4R to enhance the signaling. While the data are new and relevant to *in vivo* function of MRAP2, this study depends heavily on overexpression systems, and it is not clear whether these findings may apply *in vivo* where expression levels of receptors and signaling molecules should be much lower. Other specific points are listed below.

The reviewer addresses a very important issue that we take seriously. We agree with the reviewer that using overexpressing systems does not necessarily reflect the situation *in vivo* but we are also confident that the reviewer will appreciate that to unravel molecular mechanisms the use of heterologously expressed fluorescent fusions is an important stepping stone.

Unfortunately, up to today and to our knowledge, there is not an immortalised cell system that fully recapitulates human hypothalamic paraventricular neurons. Moreover, only one protocol exists (Wang et al. 2015) to differentiate hiPSC into arcuate nucleus neurons. In these neuronal cells both the MC3R and MC4R are expressed, both activated by the agonist α -MSH as well as

the artificial ligand NDP- α -MSH. This would make virtually impossible to discriminate between the two endogenous receptors in such cell system. Moreover, MRAP2 is also known to modulate MC3R function and is expressed together with MC4R and MC3R in the arcuate nucleus (Jamaluddin et al. 2024), making difficult to trace the effects back to the specific MC4R-MRAP2 interaction.

However, in order to address some of the shortfalls an artificial overexpressing system and move towards a more physiological setting, we already began working with murine hypothalamic cell lines. We tested three different murine hypothalamic cell lines (N7, N36 and N37) and observed that the N7 revealed a low but stable MC4R expression and therefore chose it for further experiments.

In these cells the role of MRAP2 in favouring membrane expression of heterologously expressed MC4R is more obvious than in HEK293 cells (**Supplementary Figure 6**). We have now matched these data with downstream signaling assays, showing again a marked effect of MRAP2 in affecting both Emax (as a function of MC4R receptor expressed) as well as EC50 in the cAMP production assay, with a one log-unit shift to the left in the concentration response curve, as shown in **Reviewers Figure 14**. Interestingly in N7 cells overexpressing MC4R only, the cAMP response is shifted to the right (EC50=11 nM) with respect to HEK293 cells (EC50=4.5 nM, from Figure 1), which is in agreement with decreased receptor expression on the membrane of these cells in absence of MRAP2. Moreover, the cells co-expressing MRAP2 appear to display an increased constitutive activity, again in line with our results observed in HEK293 cells. A new **Supplementary figure 14** has been added to the revised manuscript.

Reviewers Figure 14: concentration response curve of FRET ratio (donor/FRET), as a function of α MSH concentration for N7 cells transfected with MC4R, EPAC-SH187 and co-transfected with either MRAP2 (1+4) or a mock plasmid. Data are not normalised to emphasize the change in constitutive activity between the two conditions.

To address the important issue of heterologous overexpression system and the in vivo relevance we added the following paragraph to the **Discussion** section:

To investigate new functional properties of MC4R and MRAP2, the use of fluorescently tagged proteins, heterologously expressed, is the first line of experimentation required to unravel new mechanisms that might be of relevance in physiological conditions. This should be followed to investigation in autologous cell. Unfortunately, in the case of MC4R and MRAP2, this step is hampered by the lack of a human neuronal cell system that recapitulates fully the paraventricular neurons. We partially addressed this shortcoming by using a mouse hypothalamic neuronal cell line, the N7, that expresses MC4R but not MRAP2. Interestingly this allowed to show that (Supplementary Figure 6) in the absence of MRAP2, the MC4R is not expressed at the cell surface. In accordance to data obtained in the heterologous system, in N7 cells expressing MRAP2 the enhancement in Gs-mediated cAMP production observed in HEK293 cells is fully recapitulated (left shift of one order of magnitude) (Supplementary figure 14).

1. Please re-arrange supplementary figures in the order they appear in the text. It would be difficult for readers to go back and forth to navigate the supplementary figures as they read the manuscript. For example, in page 10 lines 235 and 246, supplementary figure 5D appears earlier than supplementary figure 2.

We have performed amended the manuscript as required. We shall note that some Figures (including Supplementary Figures) could be referenced again after their first introduction in the text.

2. The main text contains too many details of experimental techniques, which makes it difficult to follow the manuscript. It would be better to include only essential information here and move details to methods.

We have tried to shift as many details as possible in the Material and Methods.

3. It is currently not clear why the authors presented Figures 3B and 3C separately to explain a topic. This may be familiar to those in this field, but not to a general readership. Please resolve this issue.

This has been amended in the revised **Figure 3**, which also incorporates radioligand binding data. We further refer the reviewer to point 1 of our rebuttal to reviewer 1.

4. It may be helpful to include paragraphs explaining in vivo relevance of the findings in discussion.

We thank the reviewer for this suggestion. To address this point, we have introduced the following paragraph in the **discussion** section:

The findings reported in our study raise a few interesting aspects relevant to physiology. First and foremost, when translated at the physiological level, our results suggest that MRAP2 expression could play an important role in the modulation of MC4R-related cellular signaling. This is ultimately rooted in the co-expression of MC4R and MRAP2 in the same cells or neurons, as reported in the human brain Atlas (Supplementary Table 1).

Moreover, the recent HYPOMAP dataset confirmed co-expression of MC4R-MRAP2 as well as MC3R-MRAP2 (Jamaluddin et al. 2024), demonstrating that distinct cell populations or clusters exist where both proteins are co-expressed. Cell-type specific expression of MRAP2 emerges thus

as an additional tool that organisms can use to modulate and bias GPCR signaling in a context-dependent fashion (Bernard et al. 2023).

While until recently the potential effects of MRAP2 on MC4R signaling were considered in the context of Gs/adenylyl cyclase activation (Schonnop et al. 2016), our data together with the results from a recent study published while our manuscript was under revision (Wyatt et al. 2025), provide a complementary picture of the broad variety of signaling pathways in which MRAP2 can modulate MC4R function. These pathways include activation of Gq/11, β -arrestin recruitment, cell surface expression, receptor oligomerisation and internalization. It follows that differences in the MC4R/MRAP2 expression ratios between different tissues/cell types or in pathological conditions could have consequential impacts on the biological actions of MC4R.

Consistent with this notion, in a recent elegant study where MRAP2 was depleted from MC4R neurons (Guo et al. 2025), the authors reported that MRAP2 KO had an effect on energy metabolism by increasing hyperphagia, impaired glucose homeostasis and insulin sensitivity. In addition, it impacted the sympathetic outflow, which in turn regulates cardiovascular autonomic regulation. This study points to a more complex role of MRAP2 on MC4R physiological functions than previously anticipated; a finding that is consistent with the broad impact of MRAP2 that we observed on the cellular actions of MC4R.

Taken together, our results point to cellular basis that can underlie the role of MRAP2 in the regulation of MC4R-mediated pathways controlling metabolism, satiety and the sympathetic nervous system. Our data pave the way to test this concept for other GPCRs where signal modulation through interaction with either MRAP1 or MRAP2 has been observed (Wang et al. 2015; Chaly et al. 2016; Rouault, Buscaglia, and Sebag 2022).

Annibale, P., and M. J. Lohse. 2020. 'Spatial heterogeneity in molecular brightness', *Nat Methods*, 17: 273-75.

Bernard, A., I. Ojeda Naharros, X. Yue, F. Mifsud, A. Blake, F. Bourgain-Guglielmetti, J. Ciprin, S. Zhang, E. McDaid, K. Kim, M. V. Nachury, J. F. Reiter, and C. Vaisse. 2023. 'MRAP2 regulates energy homeostasis by promoting primary cilia localization of MC4R', *JCI Insight*, 8.

Bomberger, J. M., N. Parameswaran, C. S. Hall, N. Aiyar, and W. S. Spielman. 2005. 'Novel function for receptor activity-modifying proteins (RAMPs) in post-endocytic receptor trafficking', *J Biol Chem*, 280: 9297-307.

Brouwers, B., E. M. de Oliveira, M. Marti-Solano, F. B. F. Monteiro, S. A. Laurin, J. M. Keogh, E. Henning, R. Bounds, C. A. Daly, S. Houston, V. Ayinampudi, N. Wasiluk, D. Clarke, B. Plouffe, M. Bouvier, M. M. Babu, I. S. Farooqi, and J. Mokrosinski. 2021. 'Human MC4R variants affect endocytosis, trafficking and dimerization revealing multiple cellular mechanisms involved in weight regulation', *Cell Rep*, 34: 108862.

Chaly, A. L., D. Srisai, E. E. Gardner, and J. A. Sebag. 2016. 'The Melanocortin Receptor Accessory Protein 2 promotes food intake through inhibition of the Prokineticin Receptor-1', *Elife*, 5.

- Guo, D. F., P. A. Williams, A. Olson, D. A. Morgan, H. Herz, J. Resch, D. Atasoy, H. M. Stauss, J. A. Sebag, and K. Rahmouni. 2025. 'Loss of MRAP2 in MC4R neurons protect from obesity-associated autonomic and cardiovascular dysfunctions', *Cardiovasc Res*.
- Isbilir, A., J. Moller, M. Arimont, V. Bobkov, C. Perpina-Viciano, C. Hoffmann, A. Inoue, R. Heukers, C. de Graaf, M. J. Smit, P. Annibale, and M. J. Lohse. 2020. 'Advanced fluorescence microscopy reveals disruption of dynamic CXCR4 dimerization by subpocket-specific inverse agonists', *Proc Natl Acad Sci U S A*, 117: 29144-54.
- Isbilir, A., R. Serfling, J. Moller, R. Thomas, C. De Faveri, U. Zabel, M. Scarselli, A. G. Beck-Sickingler, A. Bock, I. Coin, M. J. Lohse, and P. Annibale. 2021. 'Determination of G-protein-coupled receptor oligomerization by molecular brightness analyses in single cells', *Nat Protoc*, 16: 1419-51.
- Jamaluddin, A., R. A. Wyatt, J. Lee, G. K. C. Dowsett, J. A. Tadross, J. Broichhagen, G. S. H. Yeo, J. Levitz, and C. M. Gorvin. 2024. 'The MRAP2 accessory protein directly interacts with melanocortin-3 receptor to enhance signaling', *bioRxiv*.
- Kleinau, G., N. A. Heyder, Y. X. Tao, and P. Scheerer. 2020. 'Structural Complexity and Plasticity of Signaling Regulation at the Melanocortin-4 Receptor', *Int J Mol Sci*, 21.
- Luo, P., W. Feng, S. Ma, A. Dai, K. Wu, X. Chen, Q. Yuan, X. Cai, D. Yang, M. W. Wang, H. Eric Xu, and Y. Jiang. 2023. 'Structural basis of signaling regulation of the human melanocortin-2 receptor by MRAP1', *Cell Res*, 33: 46-54.
- Rene, P., D. Lanfray, D. Richard, and M. Bouvier. 2021. 'Pharmacological chaperone action in humanized mouse models of MC4R-linked obesity', *JCI Insight*, 6.
- Rouault, A. A. J., P. Buscaglia, and J. A. Sebag. 2022. 'MRAP2 inhibits beta-arrestin recruitment to the ghrelin receptor by preventing GHSR1a phosphorylation', *J Biol Chem*, 298: 102057.
- Scarselli, M., P. Annibale, P. J. McCormick, S. Kolachalam, S. Aringhieri, A. Radenovic, G. U. Corsini, and R. Maggio. 2016. 'Revealing G-protein-coupled receptor oligomerization at the single-molecule level through a nanoscopic lens: methods, dynamics and biological function', *FEBS J*, 283: 1197-217.
- Schonnop, L., G. Kleinau, N. Herrfurth, A. L. Volckmar, C. Cetindag, A. Muller, T. Peters, S. Herpertz, J. Antel, J. Hebebrand, H. Biebermann, and A. Hinney. 2016. 'Decreased melanocortin-4 receptor function conferred by an infrequent variant at the human melanocortin receptor accessory protein 2 gene', *Obesity (Silver Spring)*, 24: 1976-82.
- Sebag, J. A., C. Zhang, P. M. Hinkle, A. M. Bradshaw, and R. D. Cone. 2013. 'Developmental control of the melanocortin-4 receptor by MRAP2 proteins in zebrafish', *Science*, 341: 278-81.
- Teichmann, Anke, Claudia Rutz, Annika Kreuchwig, Gerd Krause, Burkhard Wiesner, and Ralf Schüle. 2012. 'The Pseudo Signal Peptide of the Corticotropin-releasing Factor Receptor Type 2A Prevents Receptor Oligomerization', *Journal of Biological Chemistry*, 287: 27265-74.
- Uhlen, M., L. Fagerberg, B. M. Hallstrom, C. Lindskog, P. Oksvold, A. Mardinoglu, A. Sivertsson, C. Kampf, E. Sjostedt, A. Asplund, I. Olsson, K. Edlund, E. Lundberg, S. Navani, C. A. Szigartyo, J. Odeberg, D. Djureinovic, J. O. Takanen, S. Hober, T. Alm, P. H. Edqvist, H. Berling, H. Tegel, J. Mulder, J. Rockberg, P. Nilsson, J. M. Schwenk, M. Hamsten, K. von Feilitzen, M. Forsberg, L. Persson, F. Johansson, M. Zwahlen, G. von Heijne, J. Nielsen, and

- F. Ponten. 2015. 'Proteomics. Tissue-based map of the human proteome', *Science*, 347: 1260419.
- Wang, L., K. Meece, D. J. Williams, K. A. Lo, M. Zimmer, G. Heinrich, J. Martin Carli, C. A. Leduc, L. Sun, L. M. Zeltser, M. Freeby, R. Goland, S. H. Tsang, S. L. Wardlaw, D. Egli, and R. L. Leibel. 2015. 'Differentiation of hypothalamic-like neurons from human pluripotent stem cells', *J Clin Invest*, 125: 796-808.
- Ward, R. J., J. D. Padiani, S. Marsango, R. Jolly, M. R. Stoneman, G. Biener, T. M. Handel, V. Raicu, and G. Milligan. 2021. 'Chemokine receptor CXCR4 oligomerization is disrupted selectively by the antagonist ligand IT1t', *J Biol Chem*, 296: 100139.
- Webb, T. R., and A. J. Clark. 2010. 'Minireview: the melanocortin 2 receptor accessory proteins', *Mol Endocrinol*, 24: 475-84.
- Wyatt, R. A., A. Jamaluddin, V. Mistry, C. Quinn, and C. M. Gorvin. 2025. 'Obesity-associated MRAP2 variants impair multiple MC4R-mediated signaling pathways', *Hum Mol Genet*, 34: 533-46.

We are grateful to the reviewers for their positive feedback on our revised work. We have addressed any outstanding point below:

REVIEWERS' COMMENTS

Reviewer #1 (Remarks to the Author):

I am happy with the changes the authors have made to the manuscript. They have addressed most of my concerns.

Reviewer #2 (Remarks to the Author):

The author's have thoroughly responded to all our comments, and have included a large amount of new data and analyses which has further strengthened their manuscript.

Reviewer #3 (Remarks to the Author):

Reviewer #4 (Remarks to the Author):

In the revised manuscript, the authors have successfully addressed all of my previous concerns. One remaining issue is whether it is appropriate to mention that a recent study (Wyatt et al, 2025) was published while this manuscript was under revision (page 31, lines 723-724). Simply mentioning “a recent study” should be sufficient.

we thank the reviewer for this comment. We feel it is fair to acknowledge the publication of the Wyatt et al manuscript. We are open to keep the reference or change this to 'a recent study'. We defer to the Editor's advice.